



# Relevance and controls of preferential flow at the landscape scale

Dominic Demand [1], Theresa Blume [2], Markus Weiler [1]

[1] University of Freiburg, Institute of Earth and Environmental Sciences, Hydrology, Freiburg, Germany.

[2] Helmholtz Centre Potsdam, GFZ German Research Centre for Geosciences, Section Hydrology, Potsdam, Germany.

*Correspondence to*: Dominic Demand (dominic.demand@hydrology.uni-freiburg.de)

**Abstract**

The spatial and temporal controls of preferential flow (PF) during infiltration are still not fully understood. Soil moisture sensor networks give the possibility to measure infiltration response in high temporal and spatial resolution. Therefore, we

used a large-scale sensor network with 135 soil moisture profiles distributed across a complex catchment. The experimental design covers three major geological regions (Slate, Marl, Sandstone) and two land covers (forest, grassland) in Luxembourg. We analyzed the responses of up to 353 rainfall events for every of the 135 soil moisture profiles. Non-sequential responses within the soil moisture depth-profiles were taken as an indication of PF. For sequential responses wetting front velocities were determined from the observations and compared with predictions by capillary flow. A

measured wetting front velocity higher than the capillary prediction was also taken as a proxy for PF. We observed the highest fraction of non-sequential response (*NSR*) in forests on clay-rich soils (Slate, Marl). Furthermore, these two landscape units showed an increase of *NSR* with lower initial soil water content and higher maximum rainfall intensity. Wetting front velocities ranged from 6 cm day$^{-1}$ to 80640 cm day$^{-1}$ with a median of 113 cm day$^{-1}$ across all events and landscape units. The soils in the Marl geology had the highest flow velocities, independent of land cover, especially between

30 and 50 cm depth where the clay content increased. For Marl the median water content change was highest for the deepest soil moisture sensor (50 cm), whereas the other two geologies (Slate, Sandstone) showed a decrease of soil moisture change with depth. This confirms that clay content and vegetation strongly influence infiltration and reinforce preferential flow. Capillary-based soil water flow modelling was unable to predict the observed patterns. This demonstrates the danger of treating especially clay soils in the vadose zone as a low-conductivity layer, as the development of soil structure can

dominate over the effect of low-conductive texture.

## 1. Introduction

Preferential flow (PF) in soils describes different flow processes with higher flow velocities than soil matrix flow (when soil water content and soil water potential is at equilibrium) (Hendrickx and Flury, 2001). PF can affect water distribution in soil





(Ritsema et al., 1996), groundwater recharge (Ireson and Butler, 2011), root water uptake (Schwärzel et al., 2009) and solute transport (Jarvis 2016). Since the early work of Beven and Germann (1982), the importance of PF pathways such as macropores (created by roots, earthworms), fissures or cracks is widely recognized. Many studies have shown that PF is ubiquitous (Jarvis, 2007) and that "PF is the norm and not the exception" (Weiler 2017). Most of the studies focusing on

different PF processes such as fingered flow (Selker et al., 1992), macropore flow (Weiler and Naef, 2003b) or funnel flow (Kung, 1990), were carried out at the point or plot scale (spatial scale smaller than a few meters). Since PF increases the range of flow velocities in the vadose zone by magnitudes (Nimmo, 2007), it is essential to include this process when describing and modeling water and solute transport in soil. Given its importance, many models now account for PF processes (see Gerke, 2006; Köhne et al., 2009; Steinbrich et al., 2016). However, these models are difficult to apply without inverse

parameter estimation (Christiansen et al., 2004; Köhne et al., 2009) and defining meaningful parameter sets is challenging (Abbaspour et al., 2004; Arora et al., 2011; Cheng et al., 2017). Furthermore, Reck et al. (2018) showed that macropore networks and related parameters such as macropore distance and diameter are not constant over time. The problem of spatial and temporal variability of PF is also reflected in the updated paper about PF research by Beven and Germann (2013). They stated that some fundamental questions are still not solved. One of the central questions raised by the authors is: "When does

water flow through macropores in the soil?". We know about the importance of PF, but knowledge about the spatial and temporal properties affecting the distribution of PF across the landscape is still lacking (Lin et al., 2006; Wiekenkamp et al., 2016). This makes it difficult to identify hotspots or hot moments of PF.

Many methods have been developed in the last decades to study and quantify PF in soils (see e.g., Allaire et al., 2009). These methods include using X-ray tomography at the pore scale (Larsbo et al., 2014; Naveed et al., 2016), (dye) tracers (Anderson

et al., 2008; Flury et al., 1994; Zehe and Flühler, 2001a) or geophysical methods at the plot or hillslope scale (Angermann et al., 2017; Oberdörster et al., 2010). Another way to identify the potential for PF are direct measurements of the number and volume of macropores or cracks. Watson and Luxmoore (1986) used a tension infiltrometer to calculate the amount of infiltration that is caused by pores of a specific equivalent pore size, a method that has been frequently used (e.g. Buttle and McDonald, 2000). Stewart et al. (2016) measured soil crack structure and volume and used this information to model soil

water infiltration. Nevertheless, most methods lack spatial or temporal resolution to quantify the amount of PF, to derive flow velocities or water amounts at a larger scale (~km²) and the opportunity to relate them to landscape properties (such as topography, soil, land cover).

An alternative approach to study PF during infiltration are soil moisture measurements at high temporal resolution (~ minutes). While soil moisture sensors only measure at the point or profile scale, they can be deployed widely throughout the

landscape (Zehe et al., 2014). Soil moisture sensors can be installed at different depths and are minimally invasive (Hardie et al., 2013). Kim et al. (2007) and Blume et al. (2009) used soil moisture sensors to analyze infiltration responses and small-scale soil moisture patterns. Both studies found a fast soil moisture increase after rainfall events and they concluded that PF





occurred in their catchments. Lin and Zhou (2008) used soil moisture sensors for detecting PF on the catchment scale in the Shale Hills Critical Zone Observatory (Pennsylvania, USA). They defined out-of-sequence responses of the soil moisture sensors as an indication of PF. Graham and Lin (2011) and Liu and Lin (2015) further expanded the analysis of the soil moisture network to 412 rainfall events using 35 sensor profiles in the Shale Hills Critical Zone Observatory. They observed

that PF was higher when rainfall intensities were larger. PF was further sensitive to soil moisture depending on hillslope position, with higher occurrence upslope during dry conditions and downslope during wet conditions (Liu and Lin, 2015). Wiekenkamp et al. (2016) used a similar approach in the Wüstebach catchment in Germany where they studied 367 rainfall events at 101 sensor sites. They considered not only out-of-sequence responses, but also fast flow as proxy for PF. However, while the authors found that rainfall and soil moisture were important drivers, they did not observe a clear pattern with

landscape properties (topography, soil).

Using flow velocity as an indicator of PF was first established by Germann and Hensel (2006) who analyzed 100 sprinkler infiltration experiments at 25 different sites. The authors calculated wetting front velocities as the elapsed time between the first responses of two sensors at different depths along the same profile. They compared the wetting front velocities against HYDRUS-2D matrix flow simulations and found orders of magnitudes differences. Hardie et al. (2013) also applied this

method in combination with the response sequence to classify PF in an agricultural soil for 48 rainfall events in Tasmania (Australia). They found a threshold for PF with initial soil moisture but no relation to rainfall characteristics. Eguchi and Hasegawa (2008) used measured soil moisture together with one-dimensional unsaturated flow and water balance simulations to distinguish between matrix flow and PF in an Andisol.

Even though some of the study sites described above show differences in PF occurrence between soils or landscape

properties, most of them do not rigorously compare contrasting landscape units at the larger scale. Zhao et al. (2012) used the methods of Lin and Zhou (2008) for two contrasting land covers and found much higher occurrence of PF in the forest sites compared to a cropland. However, since both sites also had different soils it could not clearly be attributed to land cover. Using multiple linear regression for predicting four target variables of dye tracer flow from artificial sprinkling experiments, van Schaik (2009) found soil texture, land cover and hillslope position as important predictors of PF at a site in

Spain. Most field experiments studying the effect of soil texture and land cover on soil water flow measured infiltration characteristics or hydraulic conductivities of soil cores (Bormann and Klaassen, 2008; Gonzalez-Sosa et al., 2010; Jarvis et al., 2013; Zimmermann et al., 2006). In general, higher infiltration rates and hydraulic conductivities were observed at sites with natural vegetation or forests. These higher infiltration rates were often attributed to the presence of macropores, but not connected to the dynamics of PF occurrence under natural field conditions.

Studies linking spatial and temporal distribution of PF and soil water flow velocity with landscape attributes under natural initial and boundary conditions are still scarce. A correct estimation of PF occurrence is important for hydrological



predictions (e.g. modeling) and can improve water resource management. Therefore, the main aim of this study is to find patterns of PF at the landscape scale using profiles of soil moisture sensors distributed across contrasting soil textures, topography and land covers in a mesoscale catchment (~288 km²) under almost uniform climatic conditions. We combine soil moisture responses, flow velocities and water content changes to detect PF and to study the relevance of PF in space and

time. Furthermore, we test how well the established theory of capillary water flow (e.g. Mualem, 1976; Watson and Luxmoore, 1986) can describe the observed flow patterns. We hypothesize that PF will be the dominant process during infiltration and infiltration cannot be described by capillary theory alone. Besides initial soil moisture and rainfall characteristics, soil texture and land cover are assumed to play a major role in controlling PF. We therefore attempt to answer the following question: how important are PF contributions for different landscape units, how does this vary in time

and what are the underlying controls? Is PF temporally stable and how do the identified PF processes affect the water distribution in the vadose zone?

## 2. Material and Methods

### 2.1 Study Sites

We analyzed a dataset of 405 soil moisture sensors distributed across a complex landscape to test the hypothesis that PF dominates infiltration. The sensor sites are located in the Attert catchment in the Grand Duchy of Luxembourg. The climate is temperate semi-oceanic with a mean annual rainfall of 845 mm (Pfister et al., 2006) and mean monthly temperatures between 0°C (January) and 17°C (July) and only very few days per year with snow coverage (Wrede et al., 2015). Elevation ranges between 265 and 480 m a.s.l. and the catchment covers three major geologies (Colbach and Maquil, 2003). The

northwestern part of the catchment is located at the southern edge of the Ardennes with Devonian Slate bedrock covered by periglacial slope deposits mixed with eolian loess (Juilleret et al., 2011; Moragues-Quiroga et al., 2017). The southern part of the catchment is dominated by sedimentary rocks of the Paris Basin (Wrede et al., 2015) with Jurassic Luxembourg Sandstone at the southern catchment border and Triassic sandy Marls in the central part of the catchment (Fig.1). The Slate region has agricultural managed plateaus between steep forested slopes (~15-25°). Soil types are Haplic Cambisols (Ruptic,

Endosketelic, Siltic) (IUSS Working Group WRB, 2006) with a main texture of silty clay loam (Table 1). Texture was determined by sedimentation analysis following ISO 11277 (2002) from randomly distributed samples taken mostly in the upper 30 cm. The thickness of the Ah horizon is approximately 10 cm for forest sites and up to 30 cm for grasslands. Coarse particle fraction (> 2 mm) is estimated between 10 % and up to 50 % volume fraction in the Bw horizon and increases with depth. Layers of weathered rock (C horizon) are found usually below 50 cm. Slate rocks in the weathered layer are mostly

embedded slope parallel due to solifluction of the soil layers during the last ice age (Juilleret et al., 2011) and the bulk density of these soils is low (Wrede et al., 2015). In the Luxembourg Sandstone, Colluvic Arenosols dominate in the valley bottom and Podzols (IUSS Working Group WRB, 2006) with a sandy loam texture on the slopes and plateaus. The depth to





the unweathered bedrock is more than 2 m (Sprenger et al., 2015) with banded Bt horizons deeper than 1 m. The sandstone hillslopes are mostly forested with grasslands only present on the footslopes (Juilleret et al., 2012; Martínez-Carreras et al., 2012). The soils of the Marl geology have a more diverse texture (Wrede et al., 2015) but are often showing a clay rich layer (>50 % clay) starting between 20 and 50 cm depth. Therefore, Stagnosols (IUSS Working Group WRB, 2006) are very

common in this region. Sandy horizons are present as well, whereas topsoils mostly consist of a loamy texture. Agricultural sites and grasslands are dominant in this region with only gentle slopes (~3°). The soils show high macroporosity due to a high number of biopores and soil cracking.

In this study, the instrumentation at each site includes rainfall measurements and three soil moisture profiles separated by 5-20 meters. A soil moisture profile consists of three soil moisture sensors at 10, 30 and 50 cm depth below the surface. In total

135 soil moisture profiles at 45 different sites were distributed across the catchment (Fig. 1). The time series used in this study start between March 2012 (first installed profiles) and October 2013 (last installed profiles) and end in February 2017 (Table 1). At each of the 45 sites basic meteorological variables (temperature, humidity, radiation, wind), groundwater table elevation, sapflow, volumetric soil water content ($\theta$) and soil matrix potential were measured. The selected sites are distributed along different hillslope transects capturing different positions, slopes and aspects. The soil moisture sensors

(5TE capacitance sensors, Decagon Devices/METER Environment, USA) measured at 5-minute temporal resolution. These sensors measure with a 70 MHz frequency and have a sample volume of around 300-715 ml (Cobos, 2015; Vaz et al., 2013), although other studies found decreasing sampling volumes in wetter soils for other sensors of similar type (Blonquist et al., 2005). Due to sensor defects, 43 sensors were replaced with SMT100 (TRUEBNER GmbH, Neustadt, Germany) and 9 sensors with GS3 sensors (Decagon Devices/METER Environment, USA) in 2016. Sensors were installed horizontally with

minimum disturbance from a 30 cm diameter hole drilled with an earth drill. Each sensor was installed slightly shifted in the horizontal direction to the one above, to be unaffected by potential flow path changes by the sensor above. Furthermore, sensor cables were laid downwards in the hole first and led up on the opposite wall to prevent artificial PF along the cables leading to the sensors. In each of the three main geologies, the sensor sites were situated in two different land cover classes, forest and grassland. The selected forest sites were dominated by European beech *(Fagus sylvatica)* with occasional

occurrence of oak *(Quercus robu)*, maple *(Acer pseudoplatanus)* and spruce *(Picea abies)*. Furthermore, rainfall was measured with one tipping bucket (Davis Instruments, USA, 0.2 mm resolution) at each grassland site and five tipping buckets for capturing the spatial pattern of throughfall at each forest site. The number of soil moisture profiles for the different land cover and geological classes are summarized in Table 1. Additional information and specific site properties are shown in Appendix A. We defined six different landscape units distinguishing the three main geological formations and the

two land covers (forest, grassland) (Table 1).

Hood infiltrometer measurements (Schwärzel and Punzel, 2007) were used to determine matrix infiltration capacity. Measurements were carried out either in the direct vicinity of our sensor sites or within the same geology and land cover



class (Appendix A). Every value of matrix surface hydraulic conductivity ($K_{mat}$) consists of at least three measurement

locations, except for two sites where the infiltration rate was too high and the hood could not be filled. Hood infiltrometer

measurements were not available for grassland sites in the Luxemburg Sandstone. In total measurements from 65 locations

were used for determining $K_{mat}$ for the different landscape units. For every measurement location infiltration rates with at

least three tensions between 0.4 - 5.9 hPa were recorded to be able to fit an exponential function to calculate surface

hydraulic conductivity at a tension of 6 hPa (Gardner, 1958). At this tension, pores with a diameter ≥ 0.5 mm are excluded

from flow and measured hydraulic conductivities represent matrix infiltration capacities (Jarvis, 2007; Schwärzel and

Punzel, 2007). Hood infiltrometer measurements further gave the opportunity to estimate the macropore portion of saturated

hydraulic conductivity ($K_{MP}$). $K_{MP}$ is defined as the difference of $K_{mat}$ to the saturated hydraulic conductivity measured at a

tension of 0 hPa ($K_s$), $K_{MP} = K_s - K_{mat}$.

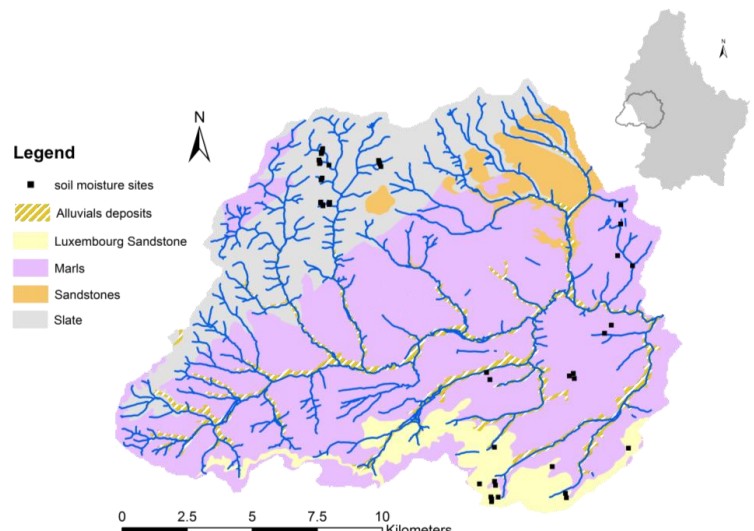

**Figure 1: Map of the Attert catchment in Luxembourg with the three main geologies and the locations of the soil moisture monitoring sites.**

**2.2. Data analysis**

**2.2.1 Event classification & soil moisture response**

Rainfall ($P$) events were defined using the 5-minute temporal resolution of the rainfall data individually for each site. For the

forest sites the mean of all five tipping buckets for every 5-minute time step was calculated to obtain throughfall. Forest

tipping buckets that measured no rainfall over one hour were excluded (assuming they were clogged), when at least three

other buckets observed rainfall during the same timeframe. Following the approach of Graham and Lin (2011) and



Wiekenkamp et al. (2016), a rainfall event was defined as rainfall with a minimum amount of 1 mm. The end was defined as the last monitored response of a rain gauge followed by a specific time period without rain ($t_e$). The sensitivity of $t_e$ on the number of rainfall events and their characteristics was investigated by testing different values of $t_e$: 3, 6, 12 and 24 consecutive hours without rain. If an event contained more than one missing value in a 2-hour period it was excluded from

further analysis. Events that were not plausible were excluded by using a threshold method for event $P$ amount (> 100 mm), average event intensity (> 15 mm h$^{-1}$) and $P$ amount in a 5-minute time step (> 6.7 mm).

Signal spikes in the measured soil moisture time series were removed by using a threshold method and data was visually checked for plausibility and consistency. In addition, sensor readings were validated against those of the other sensors in the same depth for each site. No site specific calibration of the soil moisture sensors was conducted and soil moisture values

were obtained by the sensor internal $\theta$-permittivity relationship following Topp et al. (1980). Absolute sensor accuracy of volumetric water content is ±3 Vol% (DecagonDevices, 2015). For a relative change of 1 Vol% a maximum sensor-to-sensor difference of ± 0.25 Vol% can be found in the very dry range ($\theta \sim 10$ Vol%) (Rosenbaum et al. 2010). Since Rosenbaum et al. (2011, 2012) showed that temperature effects on the sensors and on soil dielectric properties can cancel each other out, permittivity was not corrected for soil temperature. Furthermore, electrical conductivity effects of soil water on permittivity

were neglected as bulk electrical conductivity was low (< 0.1 dS m$^{-1}$) for most profiles. Although some Marl profiles show higher bulk electrical conductivities, results of soil water content change should not be affected since these profiles do not reveal fast bulk electrical conductivity fluctuations on the event scale.

For each defined rainfall event the soil moisture time series of all sensors in a profile was checked for their response. Infiltration events were defined as a $\theta$ increase of ≥1 Vol% of at least one sensor in the soil profile. This threshold was

chosen to avoid diel fluctuation being classified as events (Graham and Lin, 2011; Wiekenkamp et al., 2016). If a soil moisture event was identified, the timing of first response of every sensor was determined. The first response is defined as the point in time when the $\theta$ change is higher than the instrument noise (Lin and Zhou, 2008) that was found to be 0.4 Vol% for the 5TE sensors (Rosenbaum et al., 2010; Wiekenkamp et al., 2016). Linear interpolation was used to calculate the time between two 5 min readings to increase the temporal resolution.

The soil moisture response was tracked for up to 48 hours after the end of a rainfall event or until the time a new rainfall event starts. To have clearly separated soil water flow events that are uninfluenced by a new rainfall event for at least 24 hours, both events were excluded if a new rainfall event occurred within 24 hours after the previous event end. In the case of a response later than 24 hours we assumed that the following infiltration event is likely to be triggered by the new rainfall event (Hardie et al., 2013). Only if 99 % of the data points for all profile sensors during an infiltration event were usable,

they were considered for further analysis. To exclude snowfall or frozen soil conditions, events with a mean air temperature below 0°C during the event were excluded.



Various soil moisture and rainfall characteristics were determined for each event. Initial volumetric water content ($\theta_{ini}$) was defined as the water content before the rainfall event starts. Furthermore, change of $\theta_{ini}$ to the peak water content ($\Delta\theta_{max}$) of every event and sensor response was calculated. We grouped soil moisture into dry and wet initial conditions using $\theta$ quartiles of each profile. The total rainfall amount ($P_{sum}$), the maximum $P$ intensity in a 5-min time step ($P_{max}$) and the event

5    average rainfall intensity of the entire event ($P_{int}$) were determined. Additionally, rainfall amounts and intensities were calculated until the first soil moisture sensor response of any profile (pre-response). This was done since soil moisture classification is partly based on the first sensor response ($\theta = 0.4$ %) and later rainfall input is not further influencing the classification.

**Table 1: Site information of the six defined landscape units (* estimated by field test)**

|  | Slate | | Marl | | Sandstone | |
|---|---|---|---|---|---|---|
|  | **Forest** | **Grassland** | **Forest** | **Grassland** | **Forest** | **Grassland** |
| **No. soil moisture profiles** | 45 | 21 | 15 | 18 | 27 | 9 |
| **Dominant soil texture (USDA classification)** | silty clay loam | silty clay loam | loam (topsoil) clay* (subsoil) | clay loam (topsoil) clay (subsoil) | sandy loam | sandy loam |
| **Mean clay content [%]** | 38 | 40 | 23 / >50* (</> 30cm) | 30 / 48 (</> 30cm) | 17 | 19 |
| **Dataset period** | 03/2012-02/2017 | 04/2012-02/2017 | 03/2013-02/2017 | 09/2013-02/2017 | 03/2013-02/2017 | 07/2013-02/2017 |

### 2.2.2 Sensor response sequence and flow velocity

For all soil moisture profiles and rainfall events which met the described quality criteria, the sequence of the first sensor response was classified similar to Liu and Lin (2015) into:

(i)    no response (**NR**): none of the sensors in the profile showed a response ($\geq 1$ Vol%)

15  (ii)    sequential response (**SR**): the sensors in the profile showed a response in sequence from the uppermost sensor downwards (e.g., 10 cm to 30 cm to 50 cm or 10 cm to 30 cm). Events with only a 10 cm sensor response were also included in this group

(iii)    non-sequential response (**NSR**): events where the first response did not progress in a sequence starting from the surface (e.g., the 30 cm sensor showed a response before the 10 cm sensor)

*Non-sequential response (NSR)*

The *NSR* classification indicates a non-homogenous wetting front or bypassing of the upper soil moisture sensors, hence it is taken as a proxy for PF. *NSR* could also be a result of subsurface lateral flow or groundwater rise before the vertically

downward progressing wetting front reaches that depth (Lin and Zhou, 2008). But even in these cases, such responses describe water flow that shows either a non-homogeneous wetting front or surroundings that infiltrate water faster than the profile. Both can be seen as an indication of PF. None of the profiles showed a permanent water table in 50 cm below ground level, nevertheless some profiles are influenced by groundwater fluctuations and temporary waterlogging in 50 cm especially

during winter. The length of the time series is adequate for detecting patterns of *NSR* as Liu and Lin (2015) showed in their analysis that overall sensor response patterns show stable results using >3 years of soil moisture data.

The occurrence frequency of *NSR* was analyzed with respect to initial soil moisture, rainfall characteristics and landscape properties. All *NSR* analyses were done with pre-response rainfall characteristics. In addition, the *NSR* occurrence is compared against a theoretical capillary occurrence of PF to test the hypothesis that PF can be described by capillarity.

Classical capillary theory assumes that macropores only contribute to flow if rainfall rate exceeds the matrix infiltration capacity leading to a pore water pressure close to atmospheric pressure at the soil surface (Beven and Germann, 1982; Jarvis, 2007; Weiler, 2005). We calculated the frequency with the maximum 5-minute rainfall intensity exceeding the matrix infiltration capacity of the profile. Since matrix infiltration capacity increases under drier conditions, taking $K_{mat}$ as the matrix infiltration capacity rather underestimates this value and thus overestimates the occurrence of PF in this capillarity-

based estimation, unless soils are close to saturation.

*Statistical Analysis of NSR*

We applied a range of statistical methods to predict *NSR* occurrence and identify explanatory parameters. To describe the probability of an event to produce *NSR*, generalized linear regression models (*GLM*s) with a logistic link function were

applied (temporal *NSR* occurrence model). This was done separately for the six individual landscape units (3 geologies with 2 land covers each) and across all profiles without differentiating into landscape units. A backward stepwise model selection (stepwise AIC; software "R", package MASS) was used to reduce predictors that are either not significant or are correlated. All tested predictors can be found in Appendix A. To predict the probability of *NSR* occurrence (%) for each profile a linear model (*LM*) was fitted across all 135 profiles (spatial *NSR* occurrence model). The predictors of the *LM* were the same as for

the *GLM*, but instead of using $\theta$ and $P$ characteristics of each single event, median values across all events per profile were used.

To determine the effect of aspect on the frequency of *NSR* occurrence, only profiles at sites with slopes > 10 % were used for analysis. We distinguished between north- (270°–90°) and south-facing (90°–270°) aspects. The two-sided Wilcoxon rank sum test was used for testing of significant differences. For all other statistical comparisons in this study, a two-sided Dunn

test with Benjamini-Hochberg correction was used.

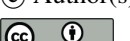


*Sequential response (SR)*

Even if an event was classified as *SR*, it cannot be excluded that PF (macropore flow, finger flow) has occurred. By comparing matrix (capillary) flow velocities to measured flow velocities, the influence of PF can be estimated. We used the approach of Germann and Hensel (2006) where the velocity of the wetting front ($v_{max}$) is defined as the velocity between the

first responses of two sensors. The upper sensor allows for the definition of a clear starting time of the water flow. Hence, vertical wetting front velocities were calculated from the *SR* for two distinct flow depths: 10 to 30 cm and 30 to 50 cm. It is important to note that $v_{max}$ represents only the fastest flow components in the sphere of influence around the soil moisture sensor (Hardie et al., 2011).

To calculate matrix flow velocity ($v_{mat}$), the 1D steady state flow equation according to Darcy's law for unsaturated

conditions was used (Hillel, 1998):

$$q = -K(\psi_m)\ \partial H / \partial z \qquad (1)$$

With $q$ being the vertical volume flux [cm day$^{-1}$], $K$ the hydraulic conductivity [cm day$^{-1}$], $\psi_m$ the matrix potential [cm], $H$ [-] the hydraulic potential including matrix potential and gravitation and $z$ the depth [cm]. For the vertical 1D case, wetting front velocity can be calculated by dividing the volume flux by the volumetric water content (Gerke, 2006):

$$v_{mat} = q / \theta \qquad (2)$$

The hydraulic gradient was calculated between two sensors using the site-specific soil water retention curves and the gravitation potential ($H = \psi_m + \psi_g$). The maximum gradient between the $\theta$ peak of the upper sensor and $\theta_{ini}$ of the lower sensor is calculated to obtain maximum $v_{mat}$. This assumes a more conservative approach since steady state assumptions are used to calculate flow velocity. Retention curves were parameterized using the van Genuchten-Mualem equation (van

Genuchten, 1980) in combination with parameter sets of Sprenger et al. (2016). The van Genuchten-Mualem parameters of Sprenger et al. (2016) do not need further corrections for matching $\theta$ with absolute values of e.g., soil core data since these parameters were calibrated for a shorter period of the same dataset. For those ten sensor sites where no parameters were determined by Sprenger et al. (2016), we simply used the mean fit for the respective geology. Although these retention parameters were inversely fitted and should therefore account for fast flow components, they rather represent matrix flow

due to the single domain Richards equation and the unimodal nature of the van Genuchten-Mualem retention function that was used (Durner et al. 1994). In addition, the fit on a daily basis does not allow for fast processes other than matrix flow. A geometric mean hydraulic conductivity was calculated between two sensors located in different depths (Zhu 2008) to obtain the effective unsaturated hydraulic conductivity of the vertical layered soil profile. The moisture content used to determine this unsaturated hydraulic conductivity was the median event water content ($\theta_{event}$), calculated from first response to the peak

water content at both sensor depths. Events that showed an upward hydraulic gradient based on this calculation were





excluded from further comparisons. Event water content was also used to calculate the matrix flow velocity ($v_{mat}$) from the volume flux ($q$).

## 3. Results

### 3.1. Rainfall and soil moisture events

The event separation method is sensitive to the required number of consecutive hours without rain ($t_e$) between the events. Table 2 shows $t_e$ values with the resulting number of events, mean event duration, rainfall amount ($P_{sum}$) and event average rainfall intensity ($P_{int}$). Shorter $t_e$ results in more events and decreasing mean event duration. Mean $P_{int}$ is gradually decreasing with longer $t_e$ due to longer event durations while mean $P_{sum}$ is increasing. We considered $t_e$ = 12 h to be sufficient to ensure event separation yielding an appropriate event length and to avoid possible superimposition of soil water flow signals from different input pulses. Therefore, the following analyses are performed with the event definition based on $t_e$ = 12 h. This results in total rainfall event numbers between 144 and 353 per profile.

**Table 2: rainfall event characteristics over all 135 profiles depending on minimum hours without rain required between consecutive rainfall events.**

|  | hours without rain ($t_e$) | | | |
| --- | --- | --- | --- | --- |
|  | **3** | **6** | **12** | **24** |
| **Sum of profile rainfall events** | 45681 | 39018 | 30207 | 18546 |
| **Mean Duration [h]** | 11.3 | 18.7 | 33.8 | 76.0 |
| **Mean $P_{sum}$ [mm]** | 5.4 | 6.4 | 8.1 | 11.9 |
| **Mean $P_{int}$ [mm h$^{-1}$]** | 0.88 | 0.65 | 0.48 | 0.33 |

Cumulative event rainfall amounts for every site are shown in Fig. 2a. The cumulative $P_{sum}$ is mainly influenced by the length of the time series and the increase with increasing number of events shows no clear difference among the six landscape units. 54.2 % of all analyzed rainfall events had sums lower than 5 mm and 77.7 % lower than 10 mm. The distribution of rainfall intensities ($P_{int}$) shows that 42.0 % of all events had a mean $P_{int} < 0.2$ mm h$^{-1}$ and 69.2 % a $P_{int} < 0.4$ mm h$^{-1}$. The density distributions show slightly higher $P_{max}$ for grassland sites but no difference among the geologies (Fig. 2b). The annual proportion of throughfall (mean annual forest $P$ / mean annual grassland $P$) varied between 62 % and 86 % for the three different geologies in the years 2014 and 2015.





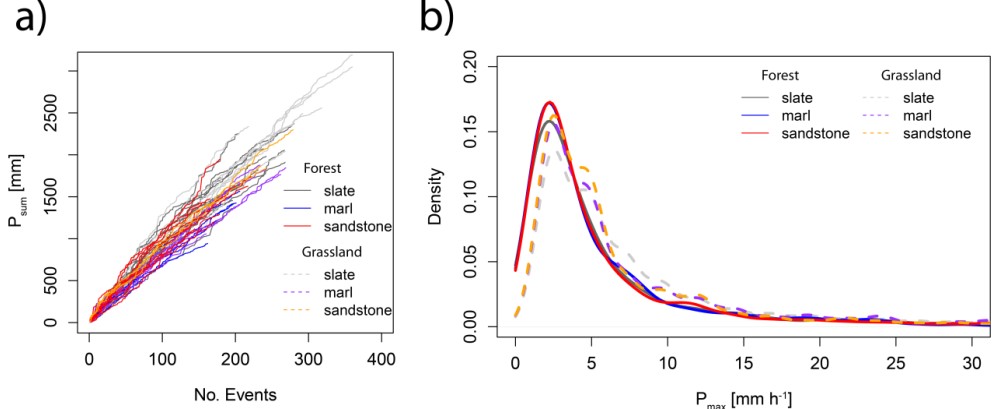

**Figure 2: Consecutive number of events vs. cumulative rainfall for each site (a) and density distribution of maximum rainfall intensity (b).**

The infiltration event response of the six defined landscape units are shown in Table 3. Between 36.2 % and 56.3 % of the rainfall events show no response (*NR*) in soil moisture, with the Marl grassland sites having the lowest amount of *NR*. 64.6 % of all *NR* events resulted from events with a $P_{sum}$ of 3 mm or less. Most detected infiltration events were of type *SR*. Under Sandstone forest sites they accounted for 55.7 %, whereas under Marl grassland sites they accounted for only 36.8 % of all events. Within the group of *SR*, 54.6 % were observed only at a depth of 10 cm, whereas sequential flow to deeper sensors occurred less frequent (21.5 % reaching 30 cm and 23.9 % 50 cm). *NSR* events were found to occur in 5.3 % to 16.1 % of all events depending on the landscape unit. The Slate and Marl forest regions showed the highest proportion (13.3 % and 16.1 %, respectively). In total 67.1 % of the *NSR* events showed a response in 30 cm first and 32.9 % in 50 cm.

**Table 3: Infiltration responses of the six landscape units.**

|  | Slate | | Marl | | Sandstone | | All |
|---|---|---|---|---|---|---|---|
|  | **Forest** | **Grassland** | **Forest** | **Grassland** | **Forest** | **Grassland** |  |
| **Sum of profile rainfall events** | 9774 | 6372 | 2823 | 4137 | 4830 | 2271 | 30207 |
| **Sum of infiltration events** | 2975 | 1121 | 733 | 852 | 1872 | 698 | 8251 |
| **NR [%]** | 43.2 | 47.2 | 36.2 | 56.3 | 39.0 | 51.9 |  |
| **SR [%]** | 43.5 | 46.1 | 47.7 | 36.8 | 55.7 | 42.8 |  |
| **NSR [%]** | 13.3 | 6.7 | 16.1 | 6.9 | 5.3 | 5.3 |  |
| **Min.-Max. profile NSR [%]** | 0 - 46.2 | 0 - 22.7 | 0 - 37.6 | 0 - 17.4 | 0- 31.8 | 0 - 15.6 |  |
| **NSR standard deviation [%]** | 9.4 | 7.5 | 11.8 | 5.4 | 8.7 | 4.8 |  |

The effect of *P* characteristics and $\theta_{ini}$ on the infiltration types were examined by calculating the median of each parameter for all infiltration events of a certain response type and their corresponding depth (Table 4). We included pre-response *P*





characteristics to show their differences between *NSR* and *SR* events. To analyze the effect of rainfall amount on infiltration depth, *SR* was also compared with the total event rainfall. *P* characteristics mainly affect the depth of the sequentially progressing soil moisture front. Response at 50 cm depth shows a median event $P_{sum}$ that is much higher than at 10 cm depth (Table 4). In addition, the $P_{max}$ is increasing with depth of response, which could partly be due to a correlation of $P_{max}$ and

$P_{sum}$ (Spearman R = 0.54). The median $\theta_{ini}$ is similar for all *SR* infiltration depths, which suggests no effect of $\theta_{ini}$ on the flow depth. Compared to the *SR* events, the median $\theta_{ini}$ of the *NSR* events is lower and also decreases with increasing depth of first response. The pre-response $P_{sum}$ is similar for *SR* and *NSR* events, while $P_{max}$ is higher for *NSR* events.

**Table 4: Rainfall characteristics of the different infiltration types and their corresponding depths (median values of all profiles and events). Sequential response (*SR*) with maximum response depth (cm) and non-sequential response**

**(*NSR*) with depth (cm) of first out-of-sequence response. Each variable was calculated for the entire event (total) and also for the time prior the first sensor response (pre-response).**

| | | SR10 | SR30 | SR50 | NSR30 | NSR50 |
|---|---|---|---|---|---|---|
| **Total** | $P_{sum}$ [mm] | 5.3 | 9.4 | 18.0 | - | - |
| | $P_{int}$ [mm h$^{-1}$] | 0.24 | 0.27 | 0.30 | - | - |
| | $P_{max}$ [mm h$^{-1}$] | 3.8 | 4.8 | 6.6 | - | - |
| **Pre-response** | $P_{sum}$ [mm] | 2.6 | 2.5 | 2.6 | 2.4 | 2.8 |
| | $P_{int}$ [mm h$^{-1}$] | 0.41 | 0.39 | 0.39 | 0.49 | 0.55 |
| | $P_{max}$ [mm h$^{-1}$] | 2.4 | 2.4 | 2.4 | 4.8 | 4.8 |
| | $\theta_{ini}$ [-] | 0.218 | 0.218 | 0.221 | 0.207 | 0.177 |

The patterns of maximum water content changes ($\Delta\theta_{max}$) in each geology were compared with respect to response type and depth. $\Delta\theta_{max}$ is taken as a proxy for the amount of water transported and can indicate differing response properties between

the geologies. Figure 3 shows $\Delta\theta_{max}$ violin plots with the *SR* at 10 and 30 cm depth also including the water content changes in the respective depth from flow events that ended at deeper sensors. For *NSR* events only $\Delta\theta_{max}$ of the first response depth was considered. Median $\Delta\theta_{max}$ values range between 1.8 and 4.3 Vol%. For the *SR* events, a significant decrease of $\Delta\theta_{max}$ with depth was observed for Slate and Sandstone sites. Marl sites did not show this damping of the water content signal with depth and exhibited a significant increase of $\Delta\theta_{max}$ at 50 cm depth (*SR*). For the NSR events no damping of $\Delta\theta_{max}$ with depth

was observed. In contrary, Sandstone and Marls both have higher $\Delta\theta_{max}$ at 50 cm depth compared to 30 cm. Furthermore, $\Delta\theta_{max}$ at 50 cm (*NSR*) was similar or higher than the least dampened response at 10 cm for SR events.





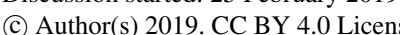

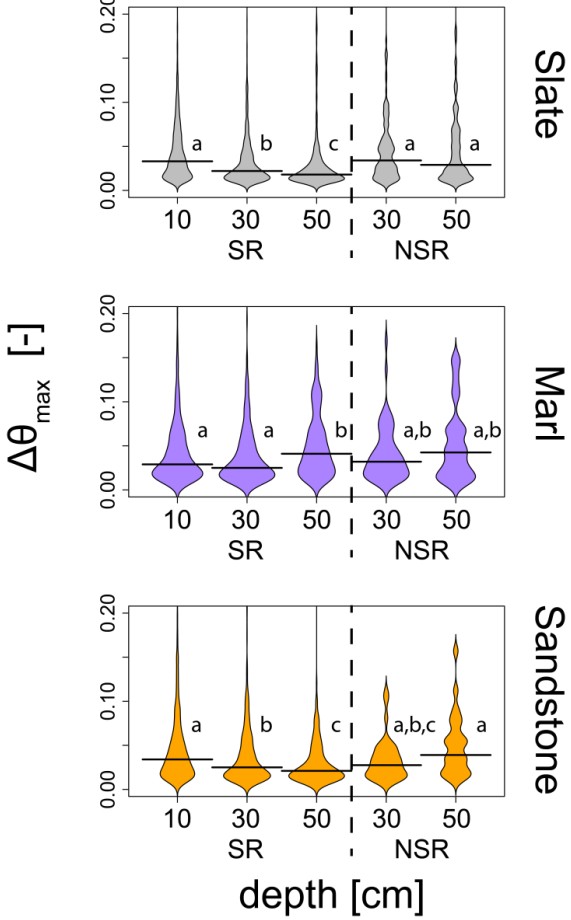

**Figure 3: Violin plots of maximum volumetric soil moisture change ($\Delta\theta_{max}$) of *SR* and *NSR* events for the three geologies differentiated by response depths. Horizontal lines in the plot indicate the median $\Delta\theta_{max}$. Same letters symbolize no significant difference between the response classes of the same geology ($p < 0.05$).**

### 3.2. Non-sequential response

*Spatial and seasonal patterns*

The fraction of *NSR* events in dependence of $\theta_{ini}$ and *P* characteristics was analyzed to reveal the spatial and temporal patterns and possible controls of PF. The single soil moisture profiles reveal a high variability with 0 to 46.2 % of the events

10   showing *NSR* (Table 3). The effect of site-specific variables on the frequency of *NSR* such as aspect, distance to the next tree and distance to stream were tested. We found no significant (p=0.819) differences between north- and south-facing aspects on the frequency of *NSR*. Furthermore, no correlation between *NSR* and distance to surrounding trees was found (Spearman



R = 0.014). None of the six individual landscape units showed a significant linear trend between *NSR* and distance to stream.

The monthly means of *NSR* across all sites show distinct seasonal dynamics (Fig. 4): From March to June *NSR* shows a

constant value of around 5 % which increases to 11.6-15.9 % from July until October and decreases again towards winter.

An increase of *NSR* with increasing $P_{max}$ was observed (Fig. 5). Especially forested sites in the Slate and Marl region showed

5    a high and significant increase of *NSR* above a threshold of $P_{max} = 10$ mm $h^{-1}$. No comparable pattern was found for the

grassland sites.

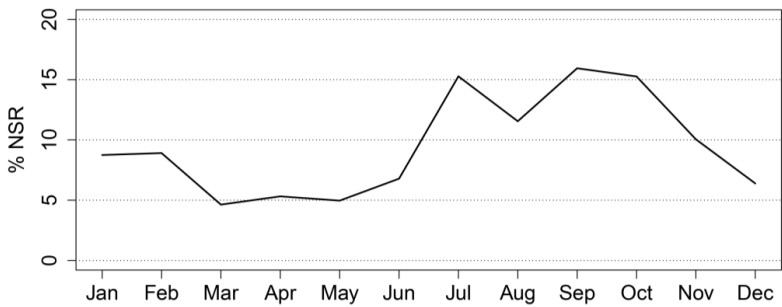

**Figure 4: Mean monthly fraction of *NSR* events over all 135 profiles.**

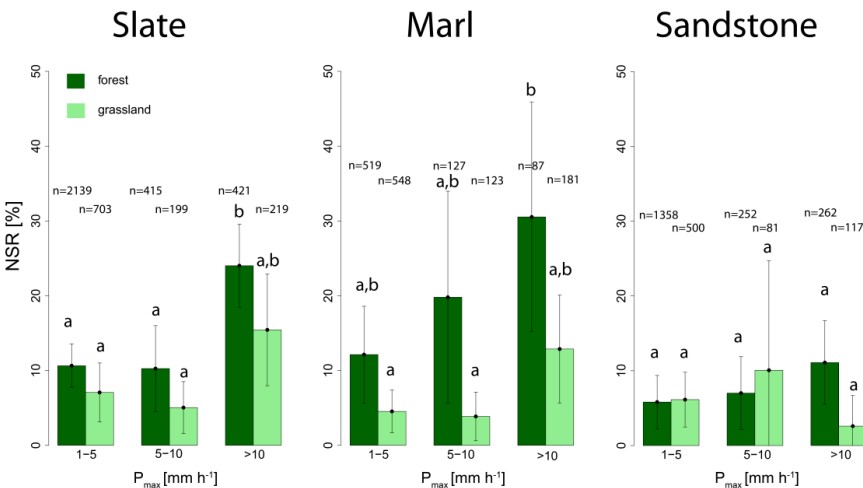

**Figure 5: *NSR* vs $P_{max}$ (pre-response). Bars indicate the mean occurrence of *NSR* and error bars show the standard**

**deviation. The numbers above the bars (*n*) indicate the total number of events. Same letters indicate no significant**

**differences of *NSR* between $P_{max}$ and land cover for each geology (*p > 0.05*).**





*Comparison of NSR observations with preferential flow from capillary prediction*

To make additional use of this unique data set, we also tested the ability of capillary theory to predict the observed *NSR* occurrence. We hypothesize that *NSR* will occur as soon as the maximum matrix infiltration capacity is reached. Thus, we compared the occurrence of *NSR* against capillary flow prediction, both in dependence of $\theta_{ini}$ for the six different landscape

5   units (Fig. 6). We observed that the drier the forested sites were, the higher the measured *NSR* occurrence was. Especially Slate and Marl sites showed a strong increase in *NSR* occurrence (of up to ~25 % of events) for the driest $\theta_{ini}$ quartile. At Slate grassland sites observed *NSR* occurrence was not related to drier conditions in the same way as for the forested sites. The fraction of *NSR* events at the Marl grassland sites did not change with initial conditions and at Sandstone grassland sites *NSR* occurrence increased only under wetter conditions. The predicted occurrence of *NSR* based on capillary theory was

10   much lower than the observed proportion of *NSR* events, except for the Marl grassland sites where measured *NSR* is lower than predicted. For some of the landscape units predicted proportion of *NSR* events was close to zero while *NSR* was actually quite frequently observed (e.g. in 24.6 % of the driest 769 soil moisture events for Slate forest). Predicted *NSR* fraction by capillary theory increases slightly under dry conditions compared to wet conditions for all landscape units due to the higher $P_{max}$ during dry summer months ($P_{max}$ = 4.8 mm h$^{-1}$ for the driest quartile and 3.36 mm h$^{-1}$ for the wettest quartile,

15   respectively). $P_{max}$ and $\theta_{ini}$ of each profile are only weakly correlated (median profile Spearman R: -0.19).

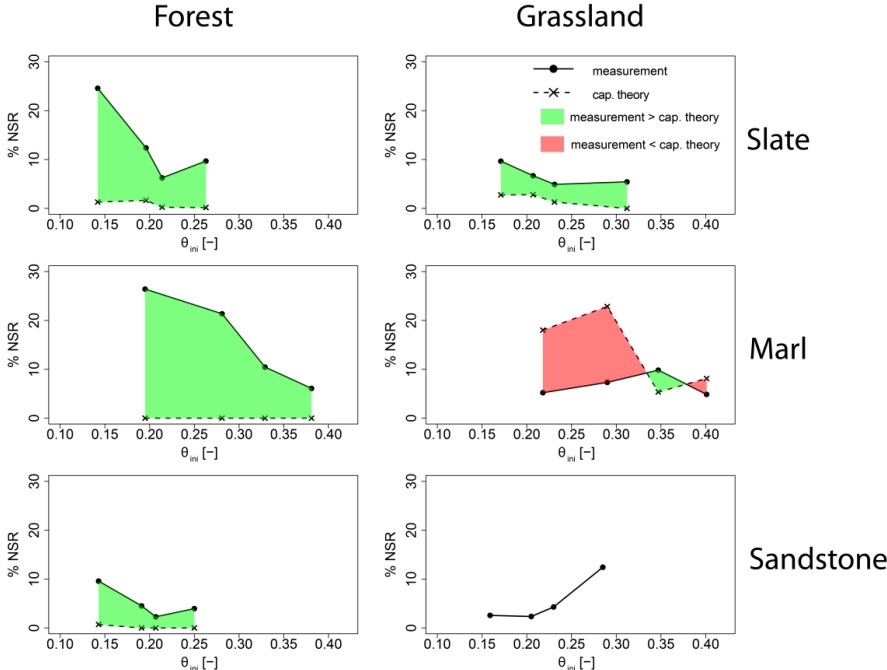

**Figure 6: Relationship of *NSR* with $\theta_{ini}$ for measured values and calculated theoretical capillary *NSR* occurrence for each geology/land cover group. Each point represents % *NSR* for all events which fall in the four different quartiles of initial soil moisture (the plotted $\theta_{ini}$ value represents a quartile median).**





*Statistical exploration: predicting preferential flow occurrence*

In a next step we used *GLM*s to predict the occurrence of *NSR* based on event and landscape characteristics (temporal occurrence). Individual models were fitted for the six landscape units (Appendix B). The stepwise model selection function

was only partly able to appropriately reduce the predictors. With our statistical modeling approach, we could again verify that the Marl and Slate forested soils are mostly influenced by initial soil moisture and *P* characteristics. Many predictors such as the hydraulic conductivities ($K_{MP}$, $K_{mat}$), distance to stream, rooting depth, $\theta_{ini}$, $P_{sum}$ and aspect were significant at the forested Sandstone sites, which exhibited the highest Pseudo-R² of all *GLM*s. Grassland sites seemed to be influenced by *P* characteristics and other landscape properties such as slope, distance to stream or elevation. All forested sites revealed a

negative relationship between $\theta_{ini}$ and probability of *NSR* events, while the grassland sites showed a positive relationship (Appendix B). One overall *GLM* was fitted without differentiating between the landscape units. Only the Sandstone grassland was excluded for the joint *GLM*, because no hood infiltrometer data was available for that group. This model produced a poor fit (McFadden Pseudo R² = 0.08) with $K_{MP}$ being the most significant predictor. Again, the other important predictors are rainfall event characteristics, $\theta_{ini}$, distance to stream, rooting depth and elevation. A linear model (spatial *NSR*

occurrence) was used to predict the proportion of *NSR* events per soil moisture profile. One model was fitted for all soil moisture profiles, excluding the Sandstone grassland sites (R²=0.40). Important predictors ($P_{sum}$, $P_{ini}$, $\theta_{ini}$, $K_{MP}$) are similar to those of the joint temporal model (*GLM*) used for all profiles (five landscape units).

**3.3 Sequential responses and flow velocities**

*Observations*

Not only *NSR* events but also *SR* events can point towards PF if the wetting front velocities are higher than expected for capillary flow in the soil matrix. The measured $v_{max}$ ranged from 6 to 80640 cm day$^{-1}$ with a median of 113 cm day$^{-1}$. Only a weak correlation was found between $v_{max}$ of the shallow versus the larger depths (10-30 cm to 30-50 cm; Spearman-R: 0.22). Median observed $v_{max}$ values per group ranged between 71 cm day$^{-1}$ for forested Sandstone sites (for the shallow depth 10-30

cm) and 297 cm day$^{-1}$ for forested Marl sites (for the depth 30-50 cm) (Fig. 7). Comparing $v_{max}$ for all landscape units the Marl soils showed more variable flow velocities and higher median values, especially between 30 and 50 cm soil depth. Slate soils do not show a significant difference between the two depths and land covers. Sandstone exhibited highest flow velocities under grassland sites. Forested Sandstone soils had a significant lower flow velocity than all other soils. Further analysis revealed no correlation between %-*NSR* and median flow velocities for each profile (Spearman R= 0.37) and $P_{max}$

and $v_{max}$ were also not significantly correlated (Spearman R = 0.22).



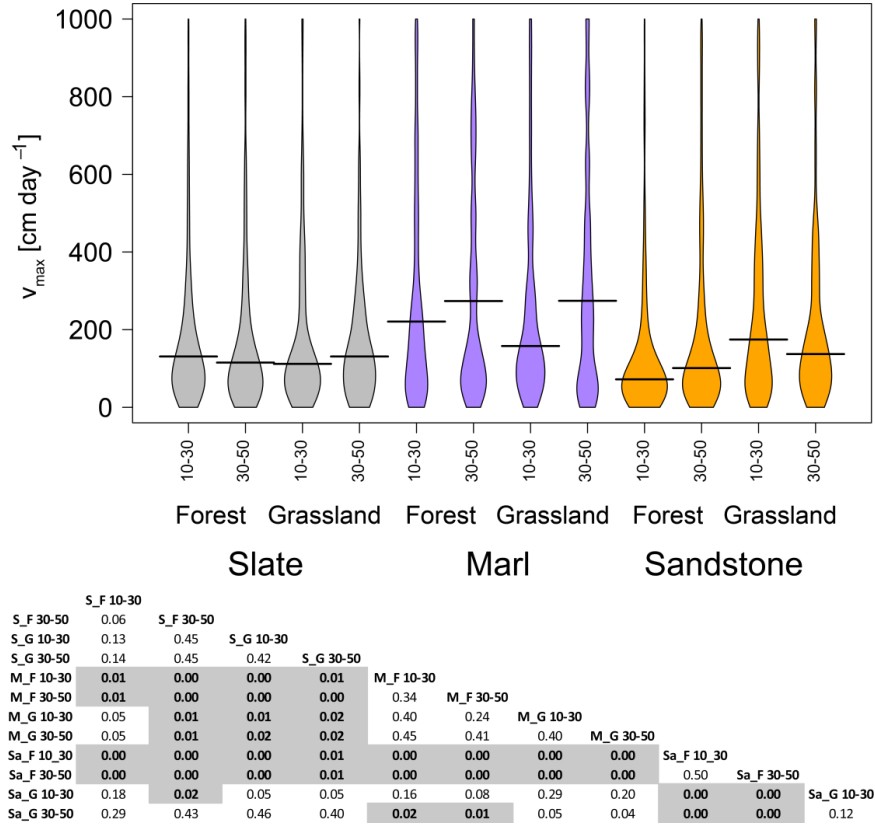

**Figure 7: Violin plot of observed $v_{max}$ for the six landscape units (colors) and two depths (10-30, 30-50 cm). The table below shows the *p*-values of the test statistics (Dunn test, two sided, Benjamini-Hochberg correction). Values above 1000 cm day$^{-1}$ are not shown. Significant differences ($p < 0.05$) are marked bold and highlighted in grey. S: Slate; M: Marl; Sa: Sandstone; F: Forest, G: Grassland and 10-30 and 30-50 are the depths between the two points where flow was measured.**

To test for a dependence of $v_{max}$ on soil water content the relationship of all observed events was shown as 2D kernel density estimations (KDE) (Venables and Ripley, 2002) with higher KDE values indicating more events (Fig. 8). The logarithmic $v$-$\theta$ relationship ($v_{max} = 10^{a\theta+b}$) was approximated by a linear regression on a semi log scale (y-axis). The correlation is significant ($p < 0.001$) and $v_{max}$ is decreasing with water content. However, the relationship only explains a small fraction of the variance (R²=0.06). $v_{max}$ showed a very high variability with fast flow velocities observed over all water contents. Furthermore, only weak $v_{max}$- $\theta$ relationships of the form $v_{max} = 10^{a\theta+b}$ were found for the individual landscape units with the highest explained variance in the Sandstone grassland between 10 and 30 cm ($R^2 = 0.17$). All landscape units showed a decrease of wetting front velocity with water content, although not all were significant.





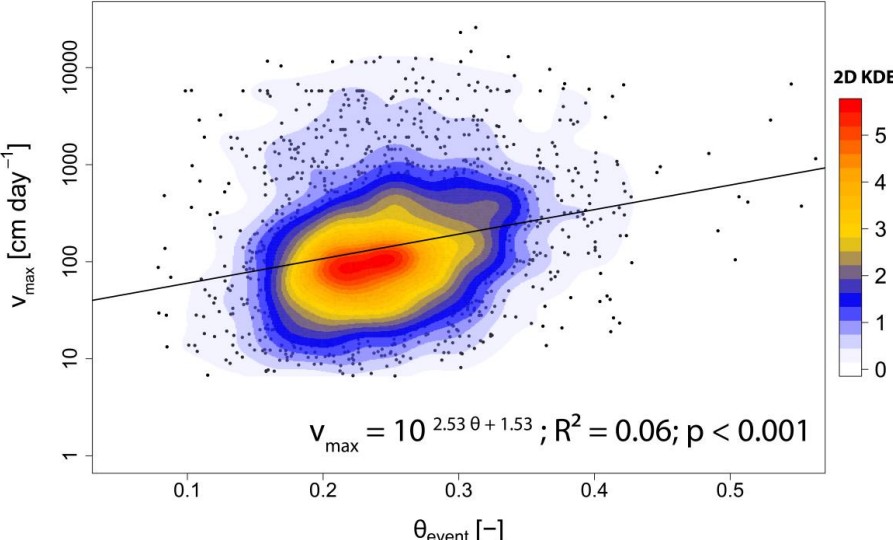

**Figure 8: Measured wetting front velocities ($v_{max}$) in relation to $\theta_{event}$. The event water content is the median water content of the two flow depths (10-30, 30-50 cm) between first response and peak soil moisture. Color contours indicate 2D kernel density estimation (2D KDE). The points show single event values. The line shows the semi-log linear fit.**

*Comparison with capillary prediction*

To identify PF from *SR* we further compared measured $v_{max}$ against calculated matrix wetting front velocities ($v_{mat}$). The relationship of measured to calculated matrix flow velocities shows a strong underestimation of $v_{max}$ by capillary matrix flow ($v_{mat}$) (Fig. 9). Wetting front velocities from capillary calculation are in part orders of magnitudes lower than the observations. Across landscape units between 84 and 96 % of the $v_{max}$ in 10-30 cm yield higher values than $v_{mat}$. Slate grassland showed with 84 % the lowest proportion of $v_{max}$ underestimated by $v_{mat}$, whereas Marl grassland shows with 96 % the highest. In 30-50 cm between 78 % (Sandstone forest) and 92 % (Marl forest) of the values have higher observed wetting front velocities than we have calculated by matrix flow.

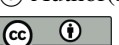

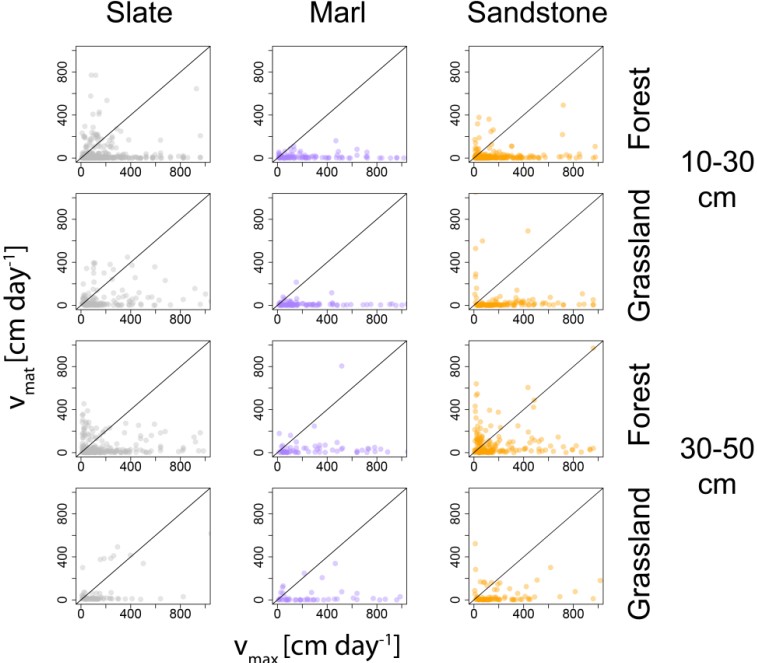

**Figure 9: Scatterplots of measured wetting front velocity ($v_{max}$) against the capillary predicted flow velocity ($v_{mat}$). The line represents the 1:1 relationship. Events with $v_{max}$ higher than 1000 cm day$^{-1}$ are not shown.**

## 4. Discussion

### 4.1 Event classification and sensor response

Dividing soil water dynamics into single events based on *P* input is always a trade-off: On the one hand, short rainfall events do not allow for a clear separation of the infiltration signals from different input pulses. On the other hand, long rainfall that is grouped into one event can result in too much information from several consecutive rain input pulses that are merged into one rainfall event. Different rainfall regimes require different threshold values, i.e. hours without rainfall ($t_e$) for the identification of event endings. While at the Shale Hills critical zone observatory a threshold value of 24 hours without rain was chosen (Liu and Lin, 2015), 12 hours seemed more appropriate in our study. However, different event definitions lead to difficulties in comparing actual numbers. This problem was already mentioned by Haas et al. (2018) in the event definition of erosion events. For an oceanic climate, with longer phases of rainfall, event-based analysis of soil water dynamics is more challenging compared to e.g. semi-arid climates with more clearly differentiable events.

*Occurrence of preferential flow*

In our study, occurrence of *NSR* for single soil moisture profiles was in the same magnitude (0-46.2 %) as those of other studies. Liu and Lin (2015) found profile *NSR* occurrence varying between <1 and 72.4 % for single years, Graham and Lin (2011) found 18 to 54 % for a three-year period and Wiekenkamp et al. (2016) found 7-51 % also using a three-year time

series. However, we found a lower average *NSR* occurrence of 5.9-14.6 % for the landscape units in our study compared to the Shale Hills catchment of Graham and Lin (2011) (26 %). Until now, most studies on *NSR* events from soil moisture time series focused on a relatively similar substrate (shale), land cover (forest) and a temperate climate (Graham and Lin, 2011; Lin and Zhou, 2008; Liu and Lin, 2015; Wiekenkamp et al., 2016). The Slate forest of our study is the landscape unit most comparable to the studies cited above. It shows a comparable range of *NSR* occurrence (max. 46.2 % for a single profile).

However, we found large differences between the landscape units in our study. Sandstone grassland showed a maximum *NSR* of only 15.6 % of the events at a single profile. Forested profiles on clayey soils (Slate & Marl) had a higher occurrence of *NSR* and a higher maximum *NSR* occurrence for single profiles within these landscape units compared to Sandstone or grassland sites. Zhao et al. (2012) also found difference in land cover (forest vs. cropland) and soil characteristics to affect *NSR* occurrence. They found lower values with 5.8 - 32.4 % *NSR* in the croplands compared to the nearby Shale Hills forest,

but also having contrasting geologies. Our study highlights the effect of land cover and geology on the occurrence of a non-homogenous wetting front by a systematical comparison. The landscape units exhibit clear patterns in *NSR*, although the variability within the landscape units is high.

**4.2 Influences on non-sequential response (*NSR*)**

We found two main properties affecting the *NSR* occurrence, the initial soil moisture ($\theta_{ini}$) and the maximum rainfall intensity ($P_{max}$) (Fig. 5 and 6). These finding are supported by the results of the *GLM* showing that $\theta_{ini}$ and $P$ characteristics were important temporal *NSR* predictors for most landscape units. However, examining the effect of $\theta_{ini}$ and $P_{max}$ in detail, results were not consistent throughout all landscape units in our study. Both effects ($\theta_{ini}$, $P_{max}$) were most strongly pronounced in the clay-rich Slate and Marl forest profiles. A higher probability of *NSR* under dryer conditions and with

higher $P$ intensities were also found by Wiekenkamp et al. (2016), Hardie et al. (2013) and Liu and Lin (2015) although they used rainfall event characteristics for the entire event instead of pre-response $P$ characteristics.

Evaluating both main factors affecting *NSR* occurrence separately, we found *NSR* decreasing with higher $\theta_{ini}$ except for Sandstone grassland. The increase of *NSR* with increasing $\theta_{ini}$ in Sandstone grassland could be an indication for macropore dominated PF with lower infiltration capacities due to higher saturation. For the other landscape units, the increase of *NSR*

with lower $\theta$ differs among the landscape units, being stronger in the Marl and Slate forest. Many studies showed that clay content increases macroporosity under dry conditions through shrinkage and the subsequent cracking of the soil (e.g. Li and





Zhang, 2011; Novák, 1999; Stewart et al., 2016a). This can lead to preferential flow as observed by dye tracers and soil moisture measurements (Hardie et al., 2011; Sander and Gerke, 2007; Zhang et al., 2014). Liu and Lin (2015) found clay content to be an important predictor of *NSR* in the Shale Hills catchment. Das Gupta et al. (2006) measured high infiltration capacity for the macropore domain of clay soils using a tension infiltrometer. The relationship between *NSR* and $\theta_{ini}$ could

also explain the observed seasonality of *NSR*, with the drier months in late summer and autumn showing the highest *NSR*. However, the question arises why the effect of clay content is much stronger in forests compared to grassland. One reason might be the higher connected macroporosity caused by roots in addition to soil cracks. Lange et al. (2009) found roots to be a key factor for preferential flow and Alaoui et al. (2011) and Gonzalez-Sosa et al. (2010) measured higher macroporosity in forest soils than for other land covers. More laterally directed PF pathways created by roots could also enhance *NSR*

(Bachmair et al., 2009). Furthermore, higher soil organic carbon content in forest can enhance aggregate stability in clayey soil and thereby be an explanation for higher *NSR* due to the resulting interaggregate porosity (Lado et al., 2004; Six et al., 2002). Carminati et al. (2009) observed root shrinkage for lupin in dry sandy soil, which might also affect the tree roots in our study sites and enhance PF in forests. Furthermore, forest are more likely to develop hydrophobic layers which can trigger PF at the soil surface (Bachmair et al., 2009; Blume et al., 2009), but this was mostly observed as finger flow on

sandy soils (Blume et al., 2009; Clothier et al., 2000; Wessolek et al., 2008). Another explanation for more *NSR* in clayey forest sites could be that forest soils become drier than grassland soils (e.g. observed by Hayati et al., 2018) due to more water uptake by trees and thus potentially more pronounced cracks (Fig. 6). While we do measure drier conditions in forests, the lack of sensor-specific calibration causes uncertainty when comparing absolute values of soil moisture.

The second factor systematically increasing the occurrence of *NSR* was higher $P_{max}$. A general effect of *P* intensity on *NSR*

can be explained by the initialization of PF: with higher *P* intensities the water pressure at the surface gets closer to zero and PF is triggered (Gjettermann et al., 1997; Jarvis, 2007; Weiler and Naef, 2003b). In our case, forest sites again showed a stronger increase in *NSR* with $P_{max}$ (in this case of throughfall). Besides the higher macroporosity that previous studies have observed in forests (see discussion above), different input fluxes could become active with higher *P* intensity. Funneling of rainfall by stemflow (not measured in this study) could be a possible mechanism enhancing PF occurrence (Schwärzel et al.,

2012). Distance to tree or rooting depth, as potential predictors of the influence of vegetation on infiltration, did not show strong correlations with *NSR* in the *GLM*s in our study. However, vegetation has numerous impacts on soil water balance and the main driver cannot clearly be determined from this analysis. The higher stone content in the Slate and the higher earthworm abundance found in the Marl, two possible drivers for PF (Bogner et al., 2008; Reck et al., 2018), were found in both land covers and thus cannot explain the higher *NSR* fraction observed in forests.

In our study, topographic properties did not seem to have a clear impact. This was also observed by Wiekenkamp et al. (2016). We neither found a difference in PF depending on aspect as observed (non-significant) by Liu and Lin (2015) nor did we find a dependence on hillslope position. Van Schaik (2009) and Zehe and Flühler (2001b) identified effects of hillslope



position on Brilliant Blue FCF infiltration patterns. However, while Zehe and Flühler (2001b) found most PF near the footslope, van Schaik (2009) found most PF dye patterns near the hilltop plateau. Liu and Lin (2015) found a temporally variable dependence on hillslope position depending on water content. This indicates that the nature of topographic controls on PF are not universal, but strongly site dependent.

That no topographic or soil properties other than clay content were found to influence infiltration processes in this and other studies can also be attributed to the heterogeneity in larger scale catchments (> km²) where two soil profiles rarely show the same overall conditions. Therefore, comparing for one effect always involves a lot of variability of other factors and singling out individual controls is difficult. For large-scale sensor networks even more soil moisture profiles than used in this study would be necessary to have enough statistical power to identify clear patterns. Consequently, in addition to soil moisture

sensor networks, it needs clear comparative studies of single properties and their interaction on soil water flow minimizing the number of other variables with experiments specifically designed for that purpose.

Comparing capillary flow theory with the measured *NSR* reveals that the occurrence of PF is underestimated by the theory most of the time. Higher occurrence of measured *NSR* compared to capillary theory prediction could indicate other initiation and flow mechanisms than pure capillary flow, such as film flow in macropores, along crack walls or in roots channels

(Germann et al., 2007; Nimmo, 2010). In addition, microtopography can be important for PF initialization and has been often ignored (Weiler and Naef, 2003b). Only in the Marl grassland much lower *NSR* is observed under dry conditions than predicted from capillarity. This is due to the low $K_{mat}$ value that has been measured in the Marl grassland and the underestimation of matrix infiltration capacity under dry conditions. However, the overestimation of *NSR* by capillary theory in the Marl grassland could also be an indication of more vertical macropore flow. The high wetting front velocities from the

*SR* reactions in this landscape unit would support the idea.

### 4.3 Flow velocities and water content

Wetting front velocities ($v_{max}$) in this study (6-80640 cm day$^{-1}$) are in the same range as other studies, however we measured slightly lower median $v_{max}$ (113 cm day$^{-1}$) than other studies (e.g. Germann and Hensel, 2006; Hardie et al., 2013; Nimmo,

2007). In addition, studies that measured $v_{max}$ in single sprinkling experiments in the Slate forest region of the Attert catchment observed higher velocities than the median $v_{max}$ found in our study: Angermann et al. (2017) measured a $v_{max}$ of 864-19000 cm day$^{-1}$ using GPR and TDR during a hillslope irrigation experiment with an intensity of 30.8 mm h$^{-1}$. Jackisch et al. (2017) observed vertical transport velocities of bromide in the range of 2732 cm day$^{-1}$ with sprinkling intensities of 30 and 50 mm h$^{-1}$. However, the highest wetting front velocity of the Slate forest landscape unit measured in our study was with

14662 cm day$^{-1}$ in a similar range.





Most of the studies mentioned above are sprinkling experiments which apply high $P$ intensities (>10 mm h$^{-1}$) and high $P_{sum}$ and thus do not provide information on the response to low intensity events that make up a large portion of the annual rainfall events (see Fig. 2). In his review, Jarvis (2007) found that solute transport studies were either carried out at (near-) saturated conditions or with high irrigation rates (> 10 mm h$^{-1}$). Langhans et al. (2011) found an increase of infiltration

capacity with higher rainfall intensity, probably due to the initiation of more macropore flow. This could be an explanation for the higher velocities found by high intensity sprinkling experiments. Therefore, a reason of partly lower $v_{max}$ observed in this study might be that we are also accounting for low $P$ intensity events. This assumption is supported by the fact that Hardie et al. (2013) measured a $v_{max}$ of 24 – 960 cm day$^{-1}$ under natural rainfall conditions which is more in the range of the results of our study. In general, it is remarkable that the high diversity of soils reported in the literature as well as in our

study produce flow velocities in a similar range independent of texture.

Comparing the variability and median of $v_{max}$ for the six landscape units, Marl profiles were most distinct to the other landscape units but did not have a significant land cover effect. Highest values were observed in 30-50 cm where most profiles show clay contents >50 %. This is in accordance to studies that showed fastest velocities due to structure development in unsaturated clay soils (Baram et al., 2012; Hardie et al., 2011; Tiktak et al., 2012). However, in the Marl also

the clay-poorer depth of 10-30 cm shows higher $v_{max}$ than the clay-rich Slate region. One reason could be the higher density of biopores that was observed in the Marl topsoils (Schneider et al., 2016). Since the amount of biopores was found to decrease with depth (Reck et al., 2018), but the velocity is increasing, clay cracks were seen as the primary influence. However, probably ponding of water on top of the clay layer and subsurface initiation of macropore flow could be a reason of higher flow velocities in the subsoil (Weiler and Naef, 2003b). Such a process was observed in the field by Hardie et al.,

(2011). This leads to the question how soil structure development in clay and flow initiation is influenced by the interaction with other factors, such as layers of contrasting texture, bulk density, stones or macropores created by flora and fauna.

In contrast to *NSR*, *SR* occurrence was more influenced by $P_{sum}$ than $\theta$ or $P_{int/max}$. Hardie et al. (2013) evaluated the rainfall characteristics for the different response types and found higher $P_{sum}$ for *SR*, although not significant in all cases. Higher $P_{sum}$ lowers the capillary forces due to higher saturation of the soil and thereby could be an explanation for the wetting front

reaching deeper sensors. This could point towards more capillary flow with *SR*. However, also macropore flow could reach deeper sensors with higher $P_{sum}$ due to less water abstraction by the matrix with a more saturated soil (Weiler and Naef, 2003a). This is the more likely explanation, since the minor effect of texture driven matrix flow during infiltration is indicated by $v_{max}$ values that are magnitudes higher than calculated capillary flow ($v_{mat}$). The $\theta$-$v_{max}$ relationship shows that even though the decrease of $v_{max}$ with $\theta$ is significant, it has little explanatory power and fast flow (>1000 cm day$^{-1}$) can

occur at any $\theta$. Furthermore, one can see orders of magnitude difference in $v_{max}$ between different events but not between the landscape units having diverse soils and land cover. A clear decline with decreasing $\theta$ comparable to unsaturated hydraulic conductivities (e.g. van Genuchten, 1980) was not observed for flow velocities under field observations. Hence, infiltration

was not primarily driven by saturation deficit. Similar results were obtained by Buttle and Turcotte (1999) who did not find a

relationship of PF and initial soil water content. Other studies (Blume et al., 2009; Hardie et al., 2011) demonstrated higher

flow velocities under dry conditions, however, this was not clear in this study. Hardie et al. (2011) found faster flow

velocities under dry conditions, which they concluded, was due to hydrophobicity and resulting finger flow. Blume et al.

(2009) measured response lags of $\theta$ in a Chilean volcanic ash soil catchment until 40 cm depth. They found response time

and thereby flow velocity to be much faster during summer time.

The different pattern of $\Delta\theta_{max}$ with depth in the Marl region (compared to Slate and Sandstone) supports the finding of

enhanced PF in the Marl. Also Hardie et al. (2013) found higher $\Delta\theta_{max}$ in greater depth during *NSR* or events with high $v_{max}$.

In combination with the higher $\Delta\theta_{max}$ during *NSR* responses this highlights the importance of PF again not only to be fast,

but also to transport significant amounts of water. This is relevant for e.g. pesticide transport, because water bypasses the

upper soil layers without much interaction.

## 5. Conclusions

Our study quantified preferential flow (PF), flow velocities and related water content changes in a heterogeneous catchment

(focusing on 3 geologies and 2 land covers). PF was found to be highly variable, but a dominant process during infiltration

that cannot be ignored. Up to 46 % of the events for single soil moisture profiles show a non-homogeneous wetting front that

often resulted in high water content changes at deeper soil zones. Furthermore, wetting front velocities of 78-96 % of the

events are underestimated by capillary flow. Our analysis revealed that high flow velocities could occur across the entire

range of soil moisture, texture and for both land covers. However, clay-rich layers showed highest median wetting front

velocities and largest water content changes. Soils with a high clay content and forest land cover had increased occurrence of

a non-homogeneous wetting front. These soils also showed a stronger dependency on initial soil moisture content (*NSR*

occurred more often under dry conditions) and maximum rainfall intensity.

Clay-rich soils in the vadose zone should not be treated as a low-conductivity layer only due to their low hydraulic

conductivity of the soil matrix. Our study highlights the importance of soil structure in clayey soil and vegetation for non-

homogenous and fast unsaturated vertical transport of water during infiltration. To account for the effect of clay and roots in

physical water flow descriptions, information on the dynamics of the soil structure in clay soils is needed as well as on root

architecture and structural interactions with the soil matrix under variable $\theta$. Further research is needed to explain the

initiation and partitioning of water into the matrix and macropore domain. We suggest to include landscape hydraulic

properties such as macropore properties rather than soil core hydraulic conductivities in large-scale physically based

hydrological models since soil cores can only partly capture the variability of complex landscapes. More effort is necessary

to find or adapt already existing approaches of measuring and monitoring PF in diverse landscapes. That includes easily

transferable relationships or pedotransfer functions, which can help to find structure-related PF parameters similar to retention parameters. We further suggest implementing large-scale sensor networks under different climatic settings, substrates, topographies, and land covers worldwide and to create standardized approaches for analyzing soil moisture datasets. Patterns identified by large-scale sensor networks need to be complemented by comparative studies on single small-

scale effects on soil water flow paths (e.g. vegetation covers or stone content). Our approach can be expanded by combining it with groundwater response time series and stable isotope methods to identify and understand flow patterns in the vadose zone at the landscape scale.

**Author contribution**

DD prepared the data, developed and performed the analysis strategy and planned and conducted the fieldwork. MW and TB designed the sensor cluster setup, were involved in their installation and contributed to the data analysis strategy. DD prepared the manuscript with contributions from all co-authors.

**Competing interests**

The authors declare that they have no conflict of interest.

**Acknowledgements**

This project was funded by the German Research Association (DFG): FOR 1598 - From Catchments as Organized Systems to Models based on Dynamic Functional Units – CAOS. Special thanks to Britta Kattenstroth, Tobias Vetter, Sibylle Hassler

and many other helpers for the installation and maintenance of the field sites. We greatly acknowledge the soil and infiltration data partly provided by Conrad Jackisch. Furthermore, we thank Natalie Orlowski for the helpful comments on the manuscript.




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



**Appendix A: Site characteristics**

**S: Slate; M: Marl; Sa: Sandstone**

**\* indicates Grassland sites**

Rooting depth was taken from Sprenger et al. (2016)

| Site ID | Elevation [m asl] | Slope [°] | Aspect [°] | Distance to stream [m] | Distance to tree profile 1 [m] | Distance to tree profile 2 [m] | Distance to tree profile 3 [m] | Rooting Depth [cm] | K_mat [cm day⁻¹] | K_MP [cm day⁻¹] | median θ profile 1 [-] | median θ profile 2 [-] | median θ profile 3 [-] |
|---|---|---|---|---|---|---|---|---|---|---|---|---|---|
| M_A | 358.2 | 4.3 | 26 | 6.0 | 3.2 | 2.4 | 4.4 | 82 | 174 | 1853 | 0.269 | 0.232 | 0.28 |
| M_B | 361.6 | 4.3 | 208 | 15.0 | 1.6 | 1.6 | 1.9 | 82 | 371 | 1218 | 0.36 | 0.336 | 0.354 |
| M_C* | 326.0 | 3.0 | 61 | 192.0 | | | | 34 | 30 | 491 | 0.28 | 0.241 | 0.226 |
| M_D* | 295.0 | 2.4 | 260 | 10.1 | | | | 30 | 30 | 491 | 0.331 | 0.37 | 0.303 |
| M_E* | 277.9 | 1.9 | 182 | 369.9 | | | | 37 | 11 | 511 | 0.344 | 0.292 | 0.398 |
| M_F* | 265.2 | 3.3 | 176 | 39.2 | | | | 63 | 11 | 511 | 0.303 | 0.359 | 0.336 |
| M_G* | 285.1 | 4.5 | 7 | 374.0 | | | | 25 | 262 | 709 | 0.364 | 0.377 | 0.337 |
| M_H | 271.3 | 3.4 | 3 | 77.0 | | | | 39 | 23 | 27 | 0.284 | 0.29 | 0.268 |
| M_I | 291.6 | 1.3 | 265 | 30.0 | 2.1 | 2.4 | 2.5 | 87 | 462 | 1184 | 0.308 | 0.291 | 0.266 |
| M_J | 282.6 | 4.6 | 244 | 6.0 | 2.1 | 2.7 | 3.6 | 99 | 499 | 733 | 0.355 | 0.381 | 0.275 |
| M_K | 282.2 | 2.9 | 173 | 10.0 | 1.3 | 4.6 | 2.8 | 99 | 462 | 1184 | 0.31 | 0.291 | 0.338 |
| S_A | 451.0 | 14.7 | 131 | 78.0 | 1 | 2.65 | 3 | 69 | 50 | 99 | 0.234 | 0.254 | 0.183 |
| S_B | 462.4 | 20.0 | 132 | 105.1 | 1.3 | 1.4 | 2.2 | 63 | 50 | 99 | 0.234 | 0.179 | 0.225 |
| S_C | 464.8 | 22.4 | 24 | 100.3 | 1 | 1.5 | 1.25 | 78 | 50 | 99 | 0.21 | 0.229 | 0.125 |
| S_D | 452.8 | 14.5 | 34 | 67.5 | 1.85 | 1.9 | 1.2 | 69 | 50 | 99 | 0.226 | 0.218 | 0.271 |
| S_E | 442.9 | 19.1 | 26 | 26.4 | 1.3 | 0.7 | 1.65 | 86 | 50 | 99 | 0.211 | 0.234 | 0.207 |
| S_F | 434.7 | 7.6 | 172 | 4.0 | 2.9 | 2.95 | 1.8 | 49 | 50 | 99 | 0.17 | 0.154 | 0.275 |
| S_G | 458.5 | 26.2 | 178 | 72.7 | 1.75 | 2.5 | 4.2 | 46 | 50 | 99 | 0.191 | 0.199 | 0.193 |





| | | | | | | | | | | | | |
|---|---|---|---|---|---|---|---|---|---|---|---|---|
| S_H | 478.0 | 10.2 | 180 | 131.8 | 1.8 | 2.5 | 2.6 | 42 | 50 | 99 | 0.202 | 0.167 | 0.193 |
| S_I* | 479.2 | 6.9 | 126 | 163.3 | | | | 35 | 57 | 80 | 0.218 | 0.21 | 0.189 |
| S_J* | 412.7 | 5.0 | 240 | 4.0 | | | | 39 | 57 | 80 | 0.378 | 0.514 | 0.229 |
| S_K* | 448.3 | 18.2 | 212 | 75.7 | | | | 35 | 57 | 80 | 0.208 | 0.228 | 0.194 |
| S_L* | 428.0 | 7.5 | 186 | 3.2 | 2.15 | 2 | 4.12 | 43 | 57 | 80 | 0.229 | 0.349 | 0.333 |
| S_M | 470.6 | 25.8 | 166 | 70.0 | 0.8 | 4.6 | 2.4 | 81 | 50 | 99 | 0.208 | 0.18 | 0.199 |
| S_O | 464.6 | 17.4 | 338 | 65.5 | | | | 107 | 50 | 99 | 0.201 | 0.241 | 0.208 |
| S_P* | 481.0 | 4.6 | 326 | 150.3 | | | | 34 | 57 | 80 | 0.203 | 0.168 | 0.203 |
| S_Q* | 453.0 | 16.8 | 183 | 35.6 | | | | 46 | 57 | 80 | 0.196 | 0.2 | 0.24 |
| S_R* | 446.4 | 13.5 | 166 | 7.7 | | | | 54 | 57 | 80 | 0.174 | 0.21 | 0.229 |
| S_S | 433.3 | 25.6 | 181 | 92.5 | 1.9 | 1.1 | 3.3 | 80 | 50 | 99 | 0.2 | 0.167 | 0.213 |
| S_T | 409.2 | 28.4 | 188 | 41.8 | 1.1 | 0.92 | 1.5 | 74 | 50 | 99 | 0.18 | 0.163 | 0.162 |
| S_U | 393.6 | 33.1 | 185 | 6.0 | 2.5 | 4.9 | 5 | 76 | 50 | 99 | 0.208 | 0.177 | 0.154 |
| S_V | 429.0 | 17.4 | 3 | 6.8 | 0.7 | 1.6 | 2.4 | 89 | 50 | 99 | 0.203 | 0.157 | 0.192 |
| S_W | 443.3 | 23.6 | 0 | 40.4 | 1.9 | 3.1 | 2.6 | 84 | 50 | 99 | 0.19 | 0.165 | 0.178 |
| Sa_A | 374.1 | 9.5 | 142 | 269.8 | 3.6 | 2.4 | 5.8 | 104 | 77 | 250 | 0.151 | 0.155 | 0.142 |
| Sa_B | 314.2 | 8.9 | 325 | 32.8 | 4.3 | 2.1 | 4.3 | 97 | 510 | 976 | 0.28 | 0.247 | 0.239 |
| Sa_C | 363.8 | 11.3 | 333 | 121.3 | 1 | 1.8 | 4.1 | 115 | 77 | 250 | 0.198 | 0.171 | 0.194 |
| Sa_D | 353.6 | 19.5 | 149 | 429.8 | 1.5 | 4.2 | 5 | 81 | 77 | 250 | 0.177 | 0.165 | 0.198 |
| Sa_E | 347.0 | 12.5 | 13 | 411.3 | 2.8 | 2.6 | 3.5 | 106 | 31 | 282 | 0.201 | 0.227 | 0.258 |
| Sa_F | 367.5 | 10.3 | 4 | 450.6 | 4.2 | 2.4 | 5 | 63 | 77 | 250 | 0.174 | 0.165 | 0.213 |
| Sa_G | 323.1 | 6.8 | 54 | 150.8 | 2.3 | 3.7 | 1.7 | 107 | 510 | 976 | 0.244 | 0.228 | 0.201 |
| Sa_H | 338.5 | 13.9 | 106 | 309.8 | 1.35 | 2.4 | 3.8 | 93 | 77 | 250 | 0.197 | 0.185 | 0.191 |
| Sa_I | 326.2 | 20.5 | 329 | 142.4 | 4.3 | 1.3 | 2.6 | 120 | 77 | 250 | 0.182 | 0.179 | 0.184 |
| Sa_J* | 297.4 | 3.7 | 323 | 104.2 | | | | 33 | | | 0.224 | 0.258 | 0.257 |
| Sa_K* | 304.9 | 10.0 | 100 | 255.4 | | | | 101 | | | 0.198 | 0.233 | 0.214 |
| Sa_L* | 297.7 | 6.8 | 300 | 256.5 | | | | 29 | | | 0.202 | 0.201 | 0.194 |




**Appendix B: Generalized linear regression models and linear model**

**Full equations for the generalized linear model (GLM) for estimating temporal occurrence for a NSR and the linear model to predict the spatial occurrence of NSR.**

GLM - Probability of event NSR

| | Intercept | $P_{sum}$ [mm] | $P_{max}$ [mm h$^{-1}$] | $P_{int}$ [mm h$^{-1}$] | $\theta_{ini}$ [-] | elevation [m a.s.l.] | aspect [°] | slope [°] | distance to stream [m] | distance to tree [m] | rooting depth [cm] | Avg. Monthly air temprature [°C] | $K_{mat}$ [cm day$^{-1}$] | $K_{MP}$ [cm day$^{-1}$] | McFadden Pseudo-R² |
|---|---|---|---|---|---|---|---|---|---|---|---|---|---|---|---|
| 5 groups | -2.452* | 0.023* | 0.025* | - | -3.851* | 0.003* | -0.001 | - | -0.003* | X | -0.007* | - | - | 0.001* | 0.08 |
| Forest — Slate | -2.784 | 0.02* | 0.036* | - | -10.97* | 0.008 | - | -0.027 | - | - | -0.005 | - | - | - | 0.09 |
| Forest — Marl | -0.538 | 0.059* | 0.042* | - | -7.172* | - | - | - | -0.062* | 0.253 | - | - | - | - | 0.18 |
| Forest — Sandstone | 4.386 | 0.035* | - | - | -14.983* | - | 0.015* | - | -0.016* | - | -0.11* | - | -0.102* | 0.062* | 0.21 |
| Grassland — Slate | -47.859* | 0.034* | 0.03* | - | 3.079 | 0.118* | - | -0.211* | -0.056* | X | -0.099 | - | - | - | 0.16 |
| Grassland — Marl | -3.351* | - | 0.051* | -0.253 | 9.078* | - | -0.012* | -0.538 | - | X | - | - | - | - | 0.10 |
| Grassland — Sandstone | 25.056 | 0.042* | - | - | - | -0.092 | - | - | - | X | - | -0.136* | X | X | 0.09 |

Linear model - % NSR on profile scale

| | Intercept | Median profile $P_{sum}$ [mm] | Median profile $P_{max}$ [mm h$^{-1}$] | Median profile $P_{int}$ [mm h$^{-1}$] | Median $\theta_{ini}$ [-] | elevation [m a.s.l.] | aspect [°] | slope [°] | distance to stream [m] | distance to tree [m] | rooting depth [cm] | Avg. Monthly air temprature [°C] | $K_{mat}$ [cm day$^{-1}$] | $K_{MP}$ [cm day$^{-1}$] | R² |
|---|---|---|---|---|---|---|---|---|---|---|---|---|---|---|---|
| 5 groups | -0.2996 | 0.0737* | - | -0.5063* | 0.5081* | 0.0004 | - | - | -0.0001 | X | -0.0009 | - | -0.0002 | 0.0001* | 0.40 |

- : Not required predictor (stepwise AIC)          * $p < 0.01$