# Peer review of "Spatio-temporal relevance and controls of preferential flow at the landscape scale"

_Hydrology and Earth System Sciences, 2019_

## Referee Comment (RC1) · Heye Bogena (Referee) · 15 Mar 2019

Heye Bogena (Referee)

h.bogena@fz-juelich.de

The authors present an interesting study in which soil moisture sensor response times after rainfall events are used characterize water preferential flow at various locations in a catchment in Luxembourg. The method relies on the sequence of soil sensor responses to rainfall events and on the water transport velocity as a means to separate infiltration events where a sequential arrival with reasonable transport speed occurs (i.e. consistent with a piston flow model) from sites with non-sequential arrival times or with unrealistically high transport speed (i.e. an indicator of preferential flow). The authors then explore different control factors which may explain the spatial and temporal occurrence of preferential flow at the catchment scale.

A better understanding of how preferential flow is generated at the catchment scale is of great interest. The methods used in this study are relatively novel (although mostly existing methods were used) and promising because they enable to investigate preferential flow events in a non-destructive manner and without the use of a non-reactive water tracer. The novelty of this study is the application of the method to different landscape units. Therefore, I think that the readers of HESS still would greatly benefit from the publication of this manuscript.

Despite the novelty and interesting approach to studying preferential flow, I think this manuscript is not yet ready for publication and would benefit from a more clear description of methods (e.g. using flow charts) and from focusing its content to the most interesting parts. The number of statistical analyses is rather excessive without providing much additional insights. For instance, the generalized linear regression model analysis is difficult to comprehend (especially for readers who are not familiar with this method) and does not provide clear results. Also the discussion section is too excessive should be focused on the most important results found in this study.

A new aspect of this study is usage of calculated of water flow velocities on basis of 1D steady state flow assumptions to identify preferential flow events. However, although the velocity-based preferential flow assessment of events with a sequential order of sensor response times is appropriate, the derived matrix flow velocities may be prone to large errors. First, the steady state assumption is violated during infiltration events and second, the hydraulic parameters derived from field data which are influenced by preferential flow. These circumstances may also explain why the measured NSR is lower for Marl grassland sites than predicted by the 1D steady state flow model. Previous sensor based preferential studies (e.g. Wiekenkamp et al., 2016) used infiltrometer measurements to assign a meaningful threshold for the matrix flow velocity. Given that fact that hood infiltrometer measurements are available for the study area I suggest to use this data to define maximum matrix flow velocities for the different landscape units.

Specific comments:

P3L5: Change to "...PF was more frequent during higher rainfall intensities."

P4L7: Change to "...and that therefore infiltration..."

P4L16: Add number of sites.

P5L5: Change to "...mostly exhibit loamy texture."

P5L5-6: Sentence reads awkward. Please reformulate and add landuse percentages.

P5L6: How was the macroporosity defined/determined?

P5L15: Change to "METER Group Inc., USA"

P6L8-9: You are actually calculating a "difference" and not a "macropore portion" of saturated hydraulic conductivity.

P7L6: Why 6.7 mm?

P7L10-11: This should be reformulated in a less unconditional way, e.g. "The manufacturer gives an accuracy of..."

P7L20: You should change "diel" into "diurnal" as this term is more common.

P7L24-25: But later you decide to use 12 hour rainfall breaks, which is somewhat confusing. In order to make this more comprehensive I suggest to present the rainfall event delineation methodology completely in the method section (thus moving P11L5-15 to the method chapter).

P8L6-8: This is not clear to me (e.g. what is the meaning of "0.4 %"?). Please be more specific.

P9L08-15: There exists a vast literature on showing that preferential flow cannot be described with classical capillary theory. Why do you still pursue this analysis although this approach is obviously prone to fail?

P9L11: The term "pore water pressure" only applies to subsurface water. Ponded water

on the soil surface shows even positive pressures.

P9L13: better: "during a rainfall event infiltration capacity decreases as the soil approaches saturation"

P9L27: Why should the aspect has an effect on the frequency of NSR occurrence?

P10L4-5: better: . . .as the velocity determined from the first responses of two sensors"

P10L9-11: During infiltration events soil water flow should be governed by nonstationary conditions. Why do you believe that your stationarity assumption can be applied?

P10L19-20: What does "in combination with parameter sets of Sprenger et al. (2016)" exactly means?

P10L23-26: Nevertheless, as the inversely derived Ks-parameter of Sprenger et al. (2016) are derived from field data, they will still be affected by preferential flow and thus will be higher compared to a Ks derived from pure matrix flow.

P11L22: Either use proportion or percentage

P12L13: Add explanations of the abbreviations

P13L18-26: This section comes somewhat out of the blue as it is not well related to the previous analysis. I suggest separating both sections and adding a short introduction to new one concerning soil water content changes.

P13L20-21: This is difficult to understand. Please try to rephrase in a more comprehensive way.

P15L2-3: This is an interesting finding. Does this correspond with seasonally varying precipitation properties?

P15L5-6: What are the possible reasons?

P16L6: The formulation "of up to ∼25 % of events" can be misinterpreted.

P16L10-11: Does this finding indicate that the inversely derived Ks-parameter of Sprenger et al. (2016) overestimate pure matrix flow?

P17L1-17: I find this section not very meaningful as it cannot be well reproduced and the results are not very much enlightening. Therefore, for the sake of comprehensibility I suggest removing it.

P18L11: It should "increasing" instead of "decreasing"

P18L15: I guess it should be again "increasing" instead of "decreasing"

P19L10: It would be interesting to see how choosing other rain gaps would influence the results (e.g. the proportion of NSR to SR events).

P21L26: Did you compare pre-response analysis with entire event analysis? This would be interesting with respect to the comparison with the other studies.

P21L28-29: I cannot follow this argument. Please explain in more detail.

P23L7-11: In my view, the results of this study rather suggest that the occurrence of preferential flow is governed by unresolved small-scale structures and processes. The study of Wiekenkamp et al. (2016) used an even denser soil moisture sensor network and still could not find landscape properties to explain their results.

P23L17-18: Why should the lower k_mat values of the Marl site lead to more NSR events and how do you know that the matrix infiltration capacity was underestimated?

P23L18-19: Why should the overestimation of NSR by capillary theory in the Marl grassland be an indication of more vertical macropore flow?

P24L23-28: These arguments are rather dubious.

P24L23-P25L11: This section is a rather excessive discussion that does not provide much additional insights.

P24L29: I guess it should be "increase" instead of "decrease".

P25L2-4: Remove repetition.

P25L16: "...showed..."

P25L17: "...in deeper..."

P25L20: "showed" instead of "had"

P25L21: You did not prove the occurrence of "non-homogeneous wetter fronts". There are other mechanisms that can lead to preferential flow (e.g. by-pass flow).

P25L23-P26L7: Please focus on presenting the main results in the conclusion section and avoid vague speculations.

Figures

Figure 9: Dots are difficult to discern.

---

## Referee Comment (RC2) · Anonymous Referee #2 · 25 Mar 2019

The manuscript by Demand et al examines the occurrence of preferential flow during infiltration for three geological regions and two land cover classes in a catchment in Luxembourg. Field measurements consisted of soil moisture measurements made at 45 sites distributed across geological and land cover classes, as well as hillslope position, slope and aspect. Each of the 45 sites consisted of three soil moisture profiles with sensors installed at three depths (10, 30 and 50 cm below the surface). Micrometeorological measurements were also collected at each site. Data were collected from 2012/2013 to 2017. Based on two proxies for preferential flow (1. non-sequential soil moisture response and 2. sequential soil moisture response velocities that exceed predicted capillary flow velocities), they found that preferential flow dominated across sites. The greatest differences between theoretical predictions of capillary flow velocities and

observed flow velocities occurred in soils with high clay content. These results suggest that preferential flow is the primary way that water infiltrates into soil in this region and that clay soils should not be considered as low hydraulic conductivity mediums.

The study tackles an important topic and contains an impressive amount of field measurements. I am a physical hydrologist, but my training and research is primarily on stream channel processes, so I am not aware of the current literature on soil infiltration and plot-scale preferential flow dynamics. Therefore, it is difficult for me to assess the novelty of the study, but it appears that the data collection and methods used are well established and that the uniqueness of this study is that they sample different geological and land cover settings for a catchment with generally uniform climate.

The introduction does a good job of reviewing the literature and setting the context for the study. My main issue with the manuscript (echoing Dr. Bogena's comments) concerns the rationale and presentation of the results. The results are very difficult to follow and it is not always clear why certain analyses were done and how they link back to the main objectives of the study. It is often not clear how and why soil profiles were grouped for certain analyses. I encourage the authors to provide more clarity on the analyses and highlight how the analyses address the research objectives.

Below, I outline two general comments, followed by some specific comments.

General comments:

a) Statistical issues

Some of the key conclusions made in this study rely on frequentist statistical testing (e.g. p-values), which, as the authors acknowledge (p23,l5-11), can be highly sensitive to sample size issues. There has also been considerable discussion recently about the major limitations to this approach (see Amrhein et al. 2019. Nature 567:305-307 for a very recent example, also Wasserstein and Lazar 2016. The American Statistician 70:129-133). It might be valuable if the authors discuss some of the inferential

uncertainties and limitations of their approach. In particular, I see three topics worth discussing:

1) pseudo-replication: It seems to me that the statistical models should be fit to the 45 sites, not the 135 soil profiles, since the grouping of three profiles within each site cannot be treated as independent. Focusing on the 135 soil profiles could be done within the GLMs if within-site variability were accounted for, but it's not clear to me that this was done.

2) sample size vs number of predictor variables used in the models: Although there is an impressive amount of data collected for this study, I'm concerned that some of the results (e.g., identification of statistically significant predictors) are simply the product of small sample sizes and noise in the model fits. For example, the GLM for Grassland-Sandstone was fit to 9 soil moisture profiles (so really, just three sites), but 13 predictor variables were used in the model fitting, which will result in an underdetermined solution. If the authors decide to keep the statistical analyses, I would suggest some sort of cross-validation exercise be done to assess the rigor of the models.

3) data exclusion. It was suggested in the methods that there is some incompleteness to the time series for each soil profile (due to logger failures and criteria for including data in the analysis). How many sites and profiles were excluded and for what time periods? This is important to know as it relates to the sample size issue outlined above.

b) Within-site and temporal variability

Instead of focusing on statistical significance, I think the authors could make an excellent contribution by focusing more on the within-site and temporal variability of their field measurements. My understanding is that the grouping of the sampling approach can be organized as: geology - land cover - site - profile. Most of the analysis focuses at the geological and land cover levels; however, throughout the manuscript I found myself constantly wanting to know more about the within-site variability in terms

of both infiltration event characteristics and soil properties. Also, at the profile level, I wanted to know more about the temporal variability. Did profiles that exhibit NSR only exhibit NSR or did they shift between NR, SR and NSR? If so, why? Instead of generalizing the results using p-values, I suggest focusing on graphical approaches to show evidence to support the research objectives.

Some specific comments:

p1,l27-28: Consider incorporating the parenthetical into the sentence - as is, this makes for a weak opening.

p2,l17: Consider removing this last sentence or expand on it to clarify to the reader what is meant by hotspots and hot moments of PF.

p2,l29-p3,l18: Consider revising these paragraphs. Right now these feel like simply a list of results from other studies. I suggest trying to better synthesize these results and identify key findings and knowledge gaps.

p4,l9-11: I think the research questions could be improved. What is meant by 'underlying controls'? Has this actually been done in this study? It seems like the PF proxies are linked to precipitation, landscape, and soil characteristics through statistical modeling. 'Underlying controls' suggests to me a more process-based approach (e.g., soil physics modelling), which isn't done in this study - outside of the predicted matrix flow velocities. What is meant by temporally stable?

p5,l25-26: What is the orifice diameter of the rainfall gauges? How was the placement of the forest gauges determined? Was variability in canopy cover and throughfall a concern?

p6,l3: Why weren't infiltrometer measurements available for the grassland/Sandstone sites?

p7,l5: Why would the sensors log these kinds of 'implausible' events? How many events were rejected because of these criteria?

p7,l29-31: How many times were data from a profile rejected because of these criteria?

Table 1: The first row highlights to me the potential issue of pseudo-replication in this study. It seems more appropriate to report the number of sites, not profiles. Also, for the soil texture and mean clay content, how variable were these values within geologic-land cover class combinations?

p9,l18-26: This paragraph is unclear. Why were some GLMs fit for individual landscape units and one GLM fit to all profiles? What was the sample size used in each of these models? If I understand this correctly, the GLM model for Sandstone-Grassland was fit to just 3 sites (9 profiles)? This seems like much too small sample size to fit models with up to 13 predictor variables.

p9,l27: Why restrict to slopes > 10%?

Fig 2a: I'm not sure what conclusions to draw from this graph? The text suggests that it shows no clear difference between landscape units; however, there appear to be considerable differences in cumulative precipitation (e.g., for event numbers $\sim$ 150, we see a range of P[sum] of almost 750 mm). Also, I don't think the various landscape units share the same event records (i.e., the x-axis is not sequential for each landscape unit), so are they comparable?

p11,l24: Why only 2014 and 2015?

p15,l1: Would a relationship between NSR and distance to stream be expected? Perhaps a provide a rationale.

Fig 4: Where do these data come from? Are these averages across all years in the study? How much did this vary between years? Does this figure account for differing number of events each year or exclusion of profiles due to logger failure or selection criteria?

Fig 8: The fit is statistically significant (but see general comment (a) above), but is this relationship practically meaningful? If you were to remove the line of best fit, I'm not

sure someone would identify a relationship in these data.

p22,l32: How much range in hillslope position was sampled? Was this the distance to stream metric?

p24,l10: Speculations on why this is?

---

## Referee Comment (RC3) · Anonymous Referee #3 · 26 Mar 2019

This study investigated preferential flow and the underlying processes on mesoscale considering various soil textures, land covers and topographic characteristics. This is done by evaluating a rich dataset of rainfall and soil moisture observations. Furthermore capillary theory is evaluated for the occurrence of non-sequential responses of soil moisture profiles. The study appears to be appropriate for HESS audience. The main novel aspect compared to preferential flow studies on local scale is the spatial and temporal coverage and inclusion of various landscape characteristics to explain variability in soil moisture profiles.

Although this study seems to potentially fill an important knowledge gap in understanding preferential flow across scales and landscapes, the manuscript in its current state is not ready for publication. Primarily the structure and selection of results should be re-

considered, but also a more defined storyline could assist the reader to extract the main novelties of this study. In general, the manuscript could benefit from reconsidering what information is necessary to broadcast the main message. I recommend to consider a few key figures that conveniently show the reader the approach and main interesting findings, instead of a long sequence of tables and graphs. Lastly, the readability would greatly increase if the authors consider a key phrase in each paragraph that, perhaps almost trivially, highlights what should be learned from the given information.

For the structure of the paper, I would recommend to consider separation of the hypothesis and throughout the paper clearly indicate which sections address information for which hypothesis. I miss this in the paper. The hypothesis could possibly be broken up in two sections. For example: 1) PF is the dominant process during infiltration, and 2) capillary theory does not suffice to explain infiltration. These can be tested for the given explanatory factors, such as land cover, geology etc. Which also gives more structure in the result and discussion section.

The generalized linear model (GLM) provided insight in the explanatory power of a large set of variables. However, as anonymous referee #2 addressed, there are some limitations to this approach. I will not re-evaluate these points, but I instead would recommend the authors to consider the use of mixed effect models. This approach allows to include random factors that potentially explain variability but are not directly incorporated in the study design. Seen the authors use R, the packages 'lme4', 'lmerTest', and 'nlme' could relatively easily allow to explore the use of mixed effect models.

On a final note, I wonder if there is any indication that the contributing area of each site is independent of the occurrence of NSR? A correlation could guide towards rising groundwater tables and associated capillary rise, or horizontal flow. Especially with high antecedent soil moisture groundwater response could be relatively fast when contributing area is large.

Since Dr. Bogena and anonymous referee #2 have readily covered a large part of my

specific comments, and technical comments seem out of place seen the current state of this manuscript, I limit myself to only the following specific comments:  

P2L17 Seems out of context to mention hotspots or hotmoments, especially as a final statement of the section. The statement needs further elaboration and references.

P2L26 '. . . scale (∼km2) and' Is this referring to 1 km2 to be considered large scale, or is a number missing?

P3L11 This section seems out of place, considering reorganizing with earlier paragraphs covering methods.

P6L1 Appendix A: consider presenting standard errors of the K measurements

P8L15 How can observations at a single depth be considered sequential?

P11L5 I would start with the most interesting finding of this study, although it could be strictly seen as a result, I could see this information to be more suited in the methods section.

P24L10 The range of reported flow velocities both in this study and other reported studies generally seems extremely large. If the range is large to begin with, how is it remarkable that they fall in the same range? Perhaps I miss a part of the reasoning.

P25L15 Awkward sentence structure

P25L29 Although this seems like an insightful comment, are there any examples how this could be implemented, or is it readily tested on small scale? A reference would be useful here.

---

## Referee Comment (RC4) · Nicholas Jarvis (Referee) · 26 Mar 2019

This paper presents and interprets a large dataset on soil water content measurements made at the landscape scale in an effort to elucidate climatic, soil and topographic controls on the occurrence of preferential flow. I think it's an interesting study which should be publishable in HESS.

As the fourth person to comment, I hesitate to add too much to what the others have already written. However, one thing that surprised me was the data on hydraulic conductivity measured by Hood infiltrometer. Some of the values at matrix saturation are as large as 500 cm/day, which seems excessively large for infiltration rates measured at a tension of 6 cm, ostensibly unaffected by soil macropores. Even some of the total

saturated hydraulic conductivities seem extraordinarily large to me, varying up to 1500 to 2000 cm/day. Maybe my surprise is just a consequence of the fact that I am more familiar with arable soils, not forest soils. No details are given of the method. I wonder, for example, how the 3D nature of the flow under the infiltrometer is accounted for? If it isn't accounted for, you could seriously overestimate K, especially in strongly layered soils. Could the authors give more details on the method?

The authors attempt to test the hypothesis that preferential flow in macropores in only generated if the rainfall rate exceeds the matrix infiltration capacity, such that the pore water pressure is close to atmospheric pressure at the soil surface. But their approach is rather indirect and therefore prone to errors and uncertainties. The best (only proper?) way to test this hypothesis would be to install tensiometers to measure soil water pressure potential, as well as the probes for soil moisture content. I think their conclusions on this point may be a little suspect, especially considering the unusually large matrix infiltration rates they measured (see above). Connected to this, I think the authors could consider re-phrasing the text at lines 739-743: these non-capillary flow mechanisms certainly contribute to flow close to saturation. However, studies of the physics of these flow processes suggests that they also require pressures quite close to atmospheric pressure for them to generate faster flow velocities than those in the matrix (see discussion and cited papers in Jarvis, 2007, p.528-529). I haven't seen any later studies that clearly contradict those findings.

The fact that preferential flow is strongest when the soil is dry suggests that the likeliest explanation of your results is the occurrence of water repellency, which is known to be a common feature of forest soils. Water repellency causes water potentials to quickly reach very close to zero, even during quite light rainfall, so that water can flow into surface-vented macropores even when the soil is dry. The authors do briefly mention hydrophobicity as a possible reason for their results (lines 694-697), but then seem to dismiss it, which I think is a pity. Preferential flow through macropores generated by the occurrence of (sub-critical) water repellency has been reported in several studies in

recent years (see those cited in the review by Jarvis et al., 2016. Vadose Zone Journal, doi:10.2136/vzj2016). I think this topic should be discussed more fully in the paper and some of these recent studies cited.

I didn't get a clear idea of whether the hypothesis on Line 145 was accepted or rejected? The first question is what is meant by "dominate"? Is it the frequency of rain events that generate preferential flow or the amount of water recharging through the unsaturated zone (or something else)? Looking at the text on lines 426 and 432-433, it would seem that preferential flow was not a dominant process (which would also tally with the very high matrix saturated hydraulic conductivities). But I got a different impression from the conclusions, at lines 828-832. Could this be clarified?

Finally, one general comment on terminology: I think it would good if the authors avoided the use of the term "wetting front" and "wetting front velocity". If you have strong preferential flow, there should not be a well-defined wetting front. Maybe you can write "maximum pore water velocity" instead of "wetting front velocity"?

Specific comments

1.) Line 41: Jarvis (2016) is not in the reference list. I think you mean Jarvis et al. 2016?

2.) Lines 64-68: you neglected one very important method and that is the analysis of breakthrough curves for non-reactive solutes (tracers). Perhaps this could be added here with one or two appropriate references?

3.) Lines 70-72: These are not really direct measurements (see line 69). In this respect, X-ray tomography of flow/transport is the only method that gives direct measurements (see Sammartino, S., et al. 2015. Identifying the functional macropore network related to preferential flow in structured soils. Vadose Zone J., doi:10.2136/vzj2015.05.0070; Koestel, J., Larsbo, M. 2014. Imaging and quantification of preferential solute transport in soil macropores. Water Resources Research, 50, 4357–4378).

4.) Line 202: robur

5.) Line 311: "non-uniform flow" is simpler and better than "non-homogeneous wetting front"

6.) Lines 324-328: This is confusing. I think it could be written more clearly and much simpler: "In addition, the hypothesis is tested that preferential flow in macropores is only generated if the rainfall rate exceeds the matrix infiltration capacity, such that the pore water pressure reaches values close to atmospheric pressure at the soil surface"

7.) Line 368: "matric" not "matrix"

8.) Line 377: Delete "Mualem"
* * *

---

## Author Comment (AC1) · 21 May 2019

Response to comments of Referee #1 Heye Bogena

We thank Heye Bogena for reviewing our manuscript and for the helpful suggestions for improving the study. We answer below to each comment in a point-by-point reply. For clarity, the comments of the referee were copied in black and our response is in blue.

**General Comments**

Despite the novelty and interesting approach to studying preferential flow, I think this manuscript is not yet ready for publication and would benefit from a more clear description of methods (e.g. using flow charts) and from focusing its content to the most interesting parts.

The number of statistical analyses is rather excessive without providing much additional insights. For instance, the generalized linear regression model analysis is difficult to comprehend (especially for readers who are not familiar with this method) and does not provide clear results.

We will add a flow chart in the method section for a clear description of the analysis. Instead of analyzing small scale spatial patterns, we will focus on the temporal dynamics of soil moisture or rainfall characteristics of large scale spatial units (landscape units). We will remove information and statistical analysis in the text that is not supporting the main findings.

We agree that the methods section and explanation of the generalized linear model (GLM) is rather short. The current GLM results give little additional insight and was also criticized by the other reviewers (also in terms of sample size and pseudo-replicates). The motivation of the GLM was to derive the probability of an event to be a preferential flow (PF) event (temporal information), additional to a linear regression model (see e.g. Liu & Lin 2015, doi:10.2136/sssaj2014.08.0330), that can provide information about the spatial PF occurrence (e.g. % PF at one location). These probabilities can be very useful for soil hydrological models to estimate not only where, but also when to include PF. Therefore, we still think that the general intention of the GLM is valuable and we will modify and simplify the temporal PF occurrence models by using a generalized linear mixed effect model (GLMM, see RC2 and RC3), only focusing on the relevant temporal predictors (initial soil moisture and rainfall). Additionally, we will explain the method, data and benefits of the model in more detail. The applied GLMM shows a good agreement with observations using the spatial information (45 sites) as a random factor. If we know the site we can estimate the probability of the event to be a PF event just

from soil moisture and rainfall data (conditional $R^2 \sim 0.7$; method of Nakagawa & Schielzeth 2013, doi: 10.1111/j.2041-210x.2012.00261.x). We think this is an interesting result since it shows that we can predict PF from temporal information if we know how the landscapes or soils behave. However, these random site factors that influence PF in the GLMM are probably small scale landscape properties at the individual sites that are unknown up to now.

Also the discussion section is too excessive should be focused on the most important results found in this study.

The result, discussion and conclusion section will be reorganized and shortened to focus on the main findings. We will also better connect the different sections to the hypothesis und thereby improve structure and readability.

A new aspect of this study is usage of calculated of water flow velocities on basis of 1D steady state flow assumptions to identify preferential flow events. However, although the velocity-based preferential flow assessment of events with a sequential order of sensor response times is appropriate, the derived matrix flow velocities may be prone to large errors. First, the steady state assumption is violated during infiltration events and second, the hydraulic parameters derived from field data which are influenced by preferential flow.

(1) Steady state assumption: During infiltration events a steady state is indeed not what we observe in nature. However, instead of a computationally intensive Richards based numerical 1D solution of all events, which also suffers from the uncertainty of parameters, the boundary conditions or discretization, we decided to use a steady state assumption of unsaturated flow. We tried to account for error that is based on the steady state assumption by using the maximum gradient during the event and are thus overestimating the driving forces. The hydraulic conductivity was previously calculated based on the median water content between the event begin and peak soil moisture. To be on the safe side, we will now change it to the maximum (peak) water content of the upper sensor. Hence, we overestimate matrix flow velocity rather than underestimating it, leading to a conservative estimate of preferential flow occurrence. Even though the absolute value is overestimated, it provides a maximum matrix flow velocity that can be used for the comparison to the magnitudes of measured flow velocities. We will clarify this in the manuscript.

(2) Parameters: The parameters will not be completely unaffected by PF. However, as we wrote in the manuscript the parameters are derived on a daily timestep over many years (including dry phases) and thus do not include soil moisture dynamics on event base (e.g. hours). This prevents optimization of the parameters to be able to account for fast flow in the range of 1000 cm/hour. Therefore, the parameters were seen as a valuable alternative to pedotransfer functions for estimating matrix flow. Additionally, the potential influence of fast flow in the estimated retention parameters of Sprenger et al. (2016) will lead again rather to an overestimate of matrix flow velocities and thus underestimate the frequency of occurrence of preferential flow.

These circumstances may also explain why the measured NSR is lower for Marl grassland sites than predicted by the 1D steady state flow model.

For the hypothesis testing of NSR ($P_{int} > K_{mat}$) in the Marl grassland the hood infiltrometer $K_{mat}$ values were used (not the predicted values by the 1D steady state flow). We apologize for the unclear description and will clarify the methodology of the different analysis in the revised manuscript.

Previous sensor based preferential studies (e.g. Wiekenkamp et al., 2016) used infiltrometer measurements to assign a meaningful threshold for the matrix flow velocity. Given that fact that hood infiltrometer measurements are available for the study area I suggest to use this data to define maximum matrix flow velocities for the different landscape units.

We will use the hood infiltrometer data on maximum matrix flow velocities as an additional method to estimate fast preferential flow and will add this to the analysis. However, using matrix saturated hydraulic conductivities for dividing into a fast/slow response only yields a binary information that results from one static threshold for all conditions. Therefore, we will also keep our analysis with the individual thresholds for every event (based on the 1D steady state flow). This allows for demonstrating the magnitude of deviation from matrix flow only and hence allows us to estimate the extent of impact of PF on the flow velocities.

**Specific comments:**

P3L5: Change to "...PF was more frequent during higher rainfall intensities.

We changed the sentence as suggested.

"P4L7: Change to "...and that therefore infiltration..."

We changed the sentence as suggested.

"P4L16: Add number of sites.

We added the number of sites to the section.

P5L5: Change to "...mostly exhibit loamy texture."

We changed the sentence as suggested.

P5L5-6: Sentence reads awkward. Please reformulate and add landuse percentages.

We reformulated the sentence and land use percentage was added.

P5L6: How was the macroporosity defined/determined?

We specified this in the text. We counted visually observable pores > 2 mm in diameter found by digging of horizontal soil profiles.

P5L15: Change to "METER Group Inc., USA"

We changed the sentence as suggested.

P6L8-9: You are actually calculating a "difference" and not a "macropore portion" of saturated hydraulic conductivity.

We agree and changed it to "macropore fraction of saturated hydraulic conductivity".

P7L6: Why 6.7 mm?

The threshold was based on an intensity measurement of 80 mm/h ≈ 6.7 mm/5 min.

P7L10-11: This should be reformulated in a less unconditional way, e.g. "The manufacturer gives an accuracy of..."

We changed the sentence as suggested.

P7L20: You should change "diel" into "diurnal" as this term is more common.

We changed the sentence as suggested.

P7L24-25: But later you decide to use 12 hour rainfall breaks, which is somewhat confusing. In order to make this more comprehensive I suggest to present the rainfall event delineation methodology completely in the method section (thus moving P11L5-15 to the method chapter).

Rainfall events were divided by 12-hour breaks, but soil moisture was tracked for additional 48 hours after the end of a rainfall event. However, we do agree that the different time steps for event definitions are confusing and will move the mentioned section (P11 L5-15 and Table 2) to the methods.

P8L6-8: This is not clear to me (e.g. what is the meaning of "0.4 %"?). Please be more specific.

We clarified the unit (0.4 Vol.%).

P9L08-15: There exists a vast literature on showing that preferential flow cannot be described with classical capillary theory. Why do you still pursue this analysis although this approach is obviously prone to fail?

Good question, however, models based on classical capillary theory are still very common. Anyhow, many people still have the misconception that preferential flow occurs only at saturation or at rainfall intensities exceeding infiltration capacity. We wanted to reiterate that this is not the case while at the same time the motivation was to see the how often classical theory fails, how strong the deviation is and if there are differences between different landscape units (probably some regions could be described by matrix flow). We will provide a clarified explanation in the revised manuscript and will further simplify the comparison of capillary theory and measurements.

 P9L11: The term "pore water pressure" only applies to subsurface water. Ponded water on the soil surface shows even positive pressures.

We agree and changed the sentence.

P9L13: better: "during a rainfall event infiltration capacity decreases as the soil approaches saturation"

We changed the sentence as suggested.

P9L27: Why should the aspect has an effect on the frequency of NSR occurrence?

Some studies found an effect of the slope aspect on the soil water properties (e.g Geroy et al. 2011, doi: 10.1002/hyp.8281). However, to focus on the main findings of our study this section was removed.

P10L4-5: better: ...as the velocity determined from the first responses of two sensors"

We changed the sentence as suggested.

P10L9-11: During infiltration events soil water flow should be governed by non-stationary conditions. Why do you believe that your stationarity assumption can be applied?

Please see our explanation under "General Comments".

P10L19-20: What does "in combination with parameter sets of Sprenger et al. (2016)" exactly means?

The formulation is confusing and was changed. van Genuchten-Mualem equation was used and parametrized with the parameters from Sprenger et al. (2016).

P10L23-26: Nevertheless, as the inversely derived Ks-parameter of Sprenger et al.(2016) are derived from field data, they will still be affected by preferential flow and thus will be higher compared to a Ks derived from pure matrix flow.

Please see our explanation under "General Comments".

P11L22: Either use proportion or percentage

We changed the sentence.

P12L13: Add explanations of the abbreviations

We added an explanation of the abbreviations.

P13L18-26: This section comes somewhat out of the blue as it is not well related to the previous analysis. I suggest separating both sections and adding a short introduction to new one concerning soil water content changes.

We separated both sections and added a short introduction as suggested.

P13L20-21: This is difficult to understand. Please try to rephrase in a more comprehensive way.

We rephrased the sentence.

P15L2-3: This is an interesting finding. Does this correspond with seasonally varying precipitation properties?

The NSR pattern corresponds to the seasonal pattern in maximum precipitation intensity (highest from June – September) and soil moisture (lowest from July – October). We will add a graph to Figure 4 showing this.

P15L5-6: What are the possible reasons?

This result is discussed on P22L21-29. Higher macroporosity, stemflow or hydrophobicity are possible reasons.

P16L6: The formulation "of up to~25 % of events" can be misinterpreted.

We changed the phrasing.

P16L10-11: Does this finding indicate that the inversely derived Ks-parameter of Sprenger et al. (2016) overestimate pure matrix flow?

Please see the explanation under "General Comments".

P17L1-17: I find this section not very meaningful as it cannot be well reproduced and the results are not very much enlightening. Therefore, for the sake of comprehensibility I suggest removing it.

Please see the explanation under "General Comments".

P18L11: It should "increasing" instead of "decreasing"

We apologized for this unclear phrasing. We meant "velocity is decreasing with decreasing water content." This was corrected.

 P18L15: I guess it should be again "increasing" instead of "decreasing"

Again we apologize for this phrasing and changed it accordingly.

P19L10: It would be interesting to see how choosing other rain gaps would influence the results (e.g. the proportion of NSR to SR events).

This was included in an earlier version of the manuscript but was removed since it made the results rather complicated and it did not add any additional information. The changes in the proportions (NSR/SR) of the reactions are relatively small (8% for Marl forest, ± 3-5% for all other landscape units), increasing with longer rain gaps for most landscape units (comparing for example 6h, 12h, and 24h rain gaps). However, there is not a clear trend of increasing proportions with longer rain gaps for all landscape units. In general the number of rain events is decreasing with longer rain gaps and events last longer (see Table 2). This leads to a decrease of soil moisture events without a reaction (NR), while SR and NSR are increasing. However, the patterns between the landscape units stay similar.

P21L26: Did you compare pre-response analysis with entire event analysis? This would be interesting with respect to the comparison with the other studies.

Similar patterns are observed using total event rainfall amount or maximum rainfall intensity of the entire event. We added a sentence to give this information for comparison. However, since the response classification is not affected by the rainfall amount or intensity after the first soil moisture sensor response we keep the pre-response in our analysis and figures.

P21L28-29: I cannot follow this argument. Please explain in more detail.

The argument will be clarified. In the sandstone grassland PF seems to be more often initialized at higher initial saturation, simply because infiltration capacity is lower and saturation is achieved faster compared to dry conditions. This is in contrast to the other landscape units with higher clay content, where more NSR is found under dry conditions with soil structure formation or hydrophobicity being the driving mechanism. We will clarify the argument.

P23L7-11: In my view, the results of this study rather suggest that the occurrence of preferential flow is governed by unresolved small-scale structures and processes. The study of Wiekenkamp et al. (2016) used an even denser soil moisture sensor network and still could not find landscape properties to explain their results.

We totally agree. However, these small-scale structures and processes can probably be attributed to landscape properties. Different combinations of the landscape properties could lead to similar flow reactions making it hard to distinguish. We hypothesize that due to the high heterogeneity of soils it would need much more sensors (even more than in Wiekenkamp et al. (2016)) to identify them. We rephrased the section.

P23L17-18: Why should the lower k_mat values of the Marl site lead to more NSR events and how do you know that the matrix infiltration capacity was underestimated?

See P9L13-15. The saturated matrix hydraulic conductivity ($K_{mat}$) was estimated using the hood infiltrometer that corresponds to the infiltration capacity at full saturation. We did not measure infiltration capacity at various moisture contents. Since the infiltration capacity is increasing with lower initial soil water content we rather underestimate the infiltration capacity under field conditions using $K_{mat}$ (because soils are rarely saturated in our catchment). P9L13-15 was moved to a different section to clarify this.

Our capillary-based estimation of NSR is again a conservative approach using this minimum infiltration capacity ($K_{mat}$). NSR is overestimated, because the infiltration capacity is

underestimated (the threshold is more often exceeded then with a higher infiltration capacity). However, to have a clearer structure of the study (especially results) and focus on the main analysis we will remove the comparison of the observed NSR responses with the estimated preferential flow reaction based on matrix hydraulic conductivity.

P23L18-19: Why should the overestimation of NSR by capillary theory in the Marl grassland be an indication of more vertical macropore flow?

We estimated more events with infiltration capacities (in our case $K_{mat}$) lower than maximum rainfall intensities in the Marl grassland and hence we should observe PF. Since we measured NSR less frequently than estimated by this approach, we hypothesized that these events have probably not resulted in PF with a NSR (non-homogenous flow), but rather in fast SR, which is supported by the high wetting front velocities. As correctly stated by the referee it has not to be vertical, but we do not observe a break in the sensor reaction sequence. We clarified the section.

P24L23-28: These arguments are rather dubious.

We agree that these arguments are rather vague and speculative, hence the section was removed.

P24L23-P25L11: This section is a rather excessive discussion that does not provide much additional insights.

We partly removed this section and some sentences were moved to a different section. In general, the discussion will be restructured and will focus on the main findings.

P24L29: I guess it should be "increase" instead of "decrease".

We apologize again for causing confusion by our repeated erroneous phrasing.

P25L2-4: Remove repetition.

We removed the repetition.

P25L16: "...showed..."

We changed the sentence as suggested.

P25L17: "...in deeper..."

We changed the sentence as suggested.

P25L20: "showed" instead of "had"

We changed the sentence as suggested.

P25L21: You did not prove the occurrence of "non-homogeneous wetter fronts". There are other mechanisms that can lead to preferential flow (e.g. by-pass flow).

We used NSR only as a proxy for preferential flow. However, we think that NSR in the first place only proved that there was a non-homogeneous wetting front that could be generated by preferential flow (see P8L22-23). This non-homogeneous wetting front can be generated by various preferential flow process (including by-pass flow). We have clarified that.

P25L23-P26L7: Please focus on presenting the main results in the conclusion section and avoid vague speculations.

We will rewrite the conclusion section and will focus on the main findings.

Figures

Figure 9: Dots are difficult to discern.

The figure will be changed.

---

## Author Comment (AC2) · 21 May 2019

**Response to comments of anonymous Referee #2**

We thank the anonymous referee #2 for reviewing our manuscript and the comments concerning mainly statistical issues. We answer below to each comment in a point-by-point reply. For clarity, the comments of the referee were copied in black and our comments are in blue.

**General Comments**

The results are very difficult to follow and it is not always clear why certain analyses were done and how they link back to the main objectives of the study. It is often not clear how and why soil profiles were grouped for certain analyses. I encourage the authors to provide more clarity on the analyses and highlight how the analyses address the research objectives.

We will add a flow chart in the method section for a clear description of the analysis and clarify the grouping of the soil moisture profiles for each analysis step. We will also better connect the different sections to the objectives and hypotheses and therefore improve structure and readability.

a) Statistical issues

Some of the key conclusions made in this study rely on frequentist statistical testing (e.g. p-values), which, as the authors acknowledge (p23,l5-11), can be highly sensitive to sample size issues. There has also been considerable discussion recently about the major limitations to this approach (see Amrhein et al. 2019. Nature 567:305-307 for a very recent example, also Wasserstein and Lazar 2016. The American Statistician 70:129-133). It might be valuable if the authors discuss some of the inferential uncertainties and limitations of their approach.

It is true that different sample sizes can lead to problems in the interpretation of test statistics (as shown in e.g. Amrhein et al. 2019. Nature 567:305-307) and that the p-value should not be used as a rigorous yes/no criterion. However, test statistics can give additional insights, especially while comparing large samples. None of our analysis and interpretations is purely based on p-value statistics and they were just added to give additional information (such as error bars, distributions etc.). Since sample size is a critical issue we added the sample size where it was missing in the manuscript to support interpretation.

1) pseudo-replication: It seems to me that the statistical models should be fit to the 45 sites, not the 135 soil profiles, since the grouping of three profiles within each site cannot be treated as independent. Focusing on the 135 soil profiles could be done within the GLMs if within-site variability were accounted for, but it's not clear to me that this was done.

The linear model (LM) was fitted to the mean NSR percentage of the 45 sites. The generalized linear model (GLM) was fitted to the infiltration events of the 135 profiles since it covers the temporal domain. Indeed pseudo-replications are an issue for the GLM. We think the general intention of the model is still important (see response RC1 Heye Bogena) and therefore modify the GLM to a generalized linear mixed effect model (GLMM) with a logit link function to account for the binomial data (NSR/no NSR) (see suggestions referee #3). By using a GLMM for the 45 sites with the individual sites as the random effect of the model we avoided pseudo-replications and treated the individual spatial landscape effects (geology, land-cover, slope, aspect etc.) together as one random landscape effect. Since the results of the GLM did not show clear landscape properties that had an effect on NSR occurrence (see RC1 Heye Bogena) we think treating it as a random factor is appropriate.

2) sample size vs number of predictor variables used in the models: Although there is an impressive amount of data collected for this study, I'm concerned that some of the results (e.g., identification of statistically significant predictors) are simply the product of small sample sizes and noise in the model fits. For example, the GLM for Grassland- Sandstone was fit to 9 soil moisture profiles (so really, just three sites), but 13 predictor variables were used in the model fitting, which will result in an underdetermined solution. If the authors decide to keep the statistical analyses, I would suggest some sort of cross-validation exercise be done to assess the rigor of the models.

The GLM was fitted to all events of the 9 profiles of the Sandstone grassland sites, which are 698 data points. Furthermore, some predictors were removed by stepwise AIC, so that only four predictors were used (not 13). Hence, the model does not result in an underdetermined solution. We will clarify that the temporal model (GLMM) was fitted to all the events of each site providing a good basis for such a statistical model.

3) data exclusion. It was suggested in the methods that there is some incompleteness to the time series for each soil profile (due to logger failures and criteria for including data in the analysis).

How many sites and profiles were excluded and for what time periods? This is important to know as it relates to the sample size issue outlined above.

We will add a diagram to the supplement that shows how many profiles in which landscape units were active or met the quality criteria over the entire time period (~2012-2017).

b) Within-site and temporal variability

Instead of focusing on statistical significance, I think the authors could make an excellent contribution by focusing more on the within-site and temporal variability of their field measurements. My understanding is that the grouping of the sampling approach can be organized as: geology - land cover - site - profile. Most of the analysis focuses at the geological and land cover levels; however, throughout the manuscript I found myself constantly wanting to know more about the within-site variability in terms of both infiltration event characteristics and soil properties. Also, at the profile level, I wanted to know more about the temporal variability. Did profiles that exhibit NSR only exhibit NSR or did they shift between NR, SR and NSR? If so, why? Instead of generalizing the results using p-values, I suggest focusing on graphical approaches to show evidence to support the research objectives.

The within-site/profile variability is indeed an interesting topic. We will add a sentence about the within-profile variability to the results. Table 3 already gives some information about the within-profile variability. However, the site or profile-level variability was not the main aim of this study. Other studies have already focused on that topic (see e.g. Wiekenkamp et al. (2016) or Liu and Lin (2015) in the reference list of the manuscript). The aim was to show the effect and variation of larger-scale landscape units with different properties. Furthermore, we wanted to identify potential temporal differences and similarities among their reactions. We will clarify the aim in the revised manuscript.

Many results in the manuscript include graphical approaches (see Fig. 3, 4, 6, 7, 8). We do not think that adding the test statistics weakens our findings (see response to General Comments).

**Specific Comments**

p1,l27-28: Consider incorporating the parenthetical into the sentence - as is, this makes for a weak opening.

We incorporated the section into the sentence.

p2,l17: Consider removing this last sentence or expand on it to clarify to the reader what is meant by hotspots and hot moments of PF.

We removed the sentence.

p2,l29-p3,l18: Consider revising these paragraphs. Right now these feel like simply a list of results from other studies. I suggest trying to better synthesize these results and identify key findings and knowledge gaps.

We will revise the paragraph and summarize the studies.

p4,l9-11: I think the research questions could be improved. What is meant by 'underlying controls'? Has this actually been done in this study? It seems like the PF proxies are linked to precipitation, landscape, and soil characteristics through statistical modeling. 'Underlying controls' suggests to me a more process-based approach (e.g., soil physics modelling), which isn't done in this study - outside of the predicted matrix flow velocities. What is meant by temporally stable?

Indeed we did not clearly identify processes. The research question will be rephrased. By "underlying controls" we meant spatial and temporal influences or drivers of preferential flow occurrence on a larger scale (e.g. landscape units).

"Temporally stable" refers to the preferential flow occurrence (if it is changing over time or not).

We will clarify both.

p5,l25-26: What is the orifice diameter of the rainfall gauges? How was the placement of the forest gauges determined? Was variability in canopy cover and throughfall a concern?

The orifice diameter of the rain gauges is 16.5 cm (collection area 214 cm²). The rain gauges were randomly placed on the 29 forest sites. The information will be added to the sentence. The experimental design of placing the five throughfall gauges aimed at covering the variability in canopy cover at each site, and variability in measured throughfall between the gauges was expected.

p6,l3: Why weren't infiltrometer measurements available for the grassland/Sandstone sites?

Hood infiltrometer measurements are often time consuming and we were not able to measure all sites during the same field campaigns.

p7,l5: Why would the sensors log these kinds of 'implausible' events? How many events were rejected because of these criteria?

During the reconnecting of the loggers following a logger error (no power etc.), the rain gauges sometimes produced this kind of implausible events. Furthermore, clogging and release of the clogged water could be a reason of the unrealistic rainfall events. Out of 10675 rainfall events (sum of rainfall events at all 45 sites), 464 rainfall events were excluded by this quality criteria. The number of rejected events was added to the sentence.

p7,l29-31: How many times were data from a profile rejected because of these criteria?

From a total of 15721 soil moisture events (sum of soil moisture event observations at all 135 soil moisture profiles), 7470 soil moisture event observations were rejected because of this criterion. If a single soil moisture sensor of a given sensor profile fails during an event, response analysis for the profile is not possible for this event and hence it is rejected. The high number of rejected events mainly results from long-term sensor failures of single sensors in different sensor profiles. Therefore, using a different quality criterion, for example 95% of usable data points of each soil moisture sensor per event, would lead to similar results (7243 soil moisture events rejected). We included the number of rejected soil moisture event and the explanation into the section.

Table 1: The first row highlights to me the potential issue of pseudo-replication in this study. It seems more appropriate to report the number of sites, not profiles. Also, for the soil texture and mean clay content, how variable were these values within geological and cover class combinations?

Table 1 is just an overview of the study sites. The different sensor responses (SR, NSR) were calculated for every single soil moisture profile. We think it is appropriate to give the number of profiles, since they determine the number of observations. We added the full textural information and the standard deviation to the supplements.

p9,l18-26: This paragraph is unclear. Why were some GLMs fit for individual landscape units and one GLM fit to all profiles? What was the sample size used in each of these models? If I understand this correctly, the GLM model for Sandstone-Grassland was fit to just 3 sites (9 profiles)? This seems like much too small sample size to fit models with up to 13 predictor variables.

Please see our response under General Comments. The GLM was fitted to the individual landscape units to test for differences in the predictors on this scale. Furthermore, we compared

it with one GLM for the whole catchment to see the potential for such an approach. We will add the sample size (number of events at each site) for the new GLMM.

p9,l27: Why restrict to slopes > 10%?

A slope < 10% is relatively flat and the orientation is not strongly pronounced. To focus on the main findings of this study (temporal variability of PF occurrence between landscape units), the analysis of the aspect was removed.

Fig 2a: I'm not sure what conclusions to draw from this graph? The text suggests that it shows no clear difference between landscape units; however, there appear to be considerable differences in cumulative precipitation (e.g., for event numbers ~ 150, we see a range of P[sum] of almost 750 mm). Also, I don't think the various landscape units share the same event records (i.e., the x-axis is not sequential for each landscape unit), so are they comparable?

It is correct that the events are not identical and necessarily sequential due to the rainfall heterogeneity, quality criteria, and the length of the time series. Therefore, single sites can show high deviation even within the same landscape unit. The motivation was to show that there is no systematic difference between the landscape units. We think that Fig. 2b) provides enough information and we will remove Fig. 2a).

p11,l24: Why only 2014 and 2015?

These were the first two years with all sites installed. Since the calculated proportions of these two years (and also the other years) do not add additional relevant information for interpretation, we removed the section to focus on the main findings (see also response to RC1).

p15,l1: Would a relationship between NSR and distance to stream be expected? Perhaps a provide a rationale.

Some authors found a relationship on hillslope position (see references P22L32-P23L4). We agree that an explanation should be mentioned earlier in the manuscript and is currently missing. However, the analysis of the small scale spatial patterns will be removed to focus on the main findings (temporal patterns of soil moisture and rainfall).

Fig 4: Where do these data come from? Are these averages across all years in the study? How much did this vary between years? Does this figure account for differing number of events each year or exclusion of profiles due to logger failure or selection criteria?

Fig. 4 is based on the same data as all other diagrams. It shows the mean NSR of all events that were measured in the twelve individual months independent of the landscape unit. Hence, the diagram averages across different years. We will modify this diagram and separate between forest and grassland sites and add additional lines to show the variability among years. The number of events used for this analysis (number of events per month for all years and the individual years) will be added to a table in the supplements.

Fig 8: The fit is statistically significant (but see general comment (a) above), but is this relationship practically meaningful? If you were to remove the line of best fit, I'm not sure someone would identify a relationship in these data.

We totally agree, the fit is statistically significant, but explains only little variation (we wrote this on P18L11-12). Furthermore, on P24L28-30 we note: "The $\theta$-$v_{max}$ relationship shows that even though the decrease of $v_{max}$ with [decreasing] $\theta$ is significant, it has little explanatory power and fast flow ($>1000$ cm day$^{-1}$) can occur at any $\theta$." We included the fit to highlight the strong variation that not simply follows the trend. We specified this in the discussion.

p22,l32: How much range in hillslope position was sampled? Was this the distance to stream metric?

The range of hillslope position was determined by the distance to stream with a range between 4 and 251 m from the different sites to the stream. Please see the table in Appendix A.

p24,l10: Speculations on why this is?

Texture seems not to be the main driver of water flow velocity during infiltration in the classical manner that fine grained texture corresponds to slow flow. Infiltration seems to be strongly controlled by PF phenomena, which are dependent on soil structure (influenced by a high clay content), biotic macropores (roots channels, earthworm borrows) and initiation processes (hydrophobicity, rain intensity). The high heterogeneity of the landscape and its temporal variation leads to PF that is caused by different drivers that are partly independent of texture (e.g. organic carbon content, number and species of soil organisms, vegetation type, rainfall characteristics). We added this as a potential explanation to the discussion.

---

## Author Comment (AC3) · 21 May 2019

Response to comments of anonymous Referee #3

We thank the anonymous referee #3 for reviewing our manuscript and his suggestion for improving the temporal occurrence model of preferential flow (PF). We answer below to each comment in a point-by-point reply. For clarity, the comments of the referee were copied in black and our comments are in blue.

**General Comments**

Primarily the structure and selection of results should be reconsidered, but also a more defined storyline could assist the reader to extract the main novelties of this study. In general, the manuscript could benefit from reconsidering what information is necessary to broadcast the main message. I recommend to consider a few key figures that conveniently show the reader the approach and main interesting findings, instead of a long sequence of tables and graphs. Lastly, the readability would greatly increase if the authors consider a key phrase in each paragraph that, perhaps almost trivially, highlights what should be learned from the given information.

For the structure of the paper, I would recommend to consider separation of the hypothesis and throughout the paper clearly indicate which sections address information for which hypothesis. I miss this in the paper. The hypothesis could possibly be broken up in two sections. For example: 1) PF is the dominant process during infiltration, and 2) capillary theory does not suffice to explain infiltration. These can be tested for the given explanatory factors, such as land cover, geology etc. Which also gives more structure in the result and discussion section.

We will restructure the results and focus on the main findings, the temporal dynamics of PF in different large scale spatial units (landscape units). By removing most of the small-scale spatial analysis we will highlight the main storyline. Additionally, we will add introduction phrases and summaries for the single sections. We will clearly address which section of the results and discussion contributes to which hypothesis.

The generalized linear model (GLM) provided insight in the explanatory power of a large set of variables. However, as anonymous referee #2 addressed, there are some limitations to this approach. I will not re-evaluate these points, but I instead would recommend the authors to

consider the use of mixed effect models. This approach allows to include random factors that potentially explain variability but are not directly incorporated in the study design. Seen the authors use R, the packages 'lme4', 'lmerTest', and 'nlme' could relatively easily allow to explore the use of mixed effect models.

We thank you for the suggestion and have changed the GLM to a generalized linear mixed effect model (GLMM) that incorporates the spatial site information as a random effect (see response to RC1 and RC2).

On a final note, I wonder if there is any indication that the contributing area of each site is independent of the occurrence of NSR? A correlation could guide towards rising groundwater tables and associated capillary rise, or horizontal flow. Especially with high antecedent soil moisture groundwater response could be relatively fast when contributing area is large.

We have calculated the upslope contributing area for each site and compared it against NSR occurrence. The Spearman R is 0.1 and hence, influence of groundwater in the upper 0.5 m of soil seems to be small for our sites. This is supported by the fact that % NSR and distance to stream does not show a correlation (P15L1).

**Specific Comments**

P2L17 Seems out of context to mention hotspots or hot moments, especially as a final statement of the section. The statement needs further elaboration and references.

We agree that the sentence is out of context at the end of this section and removed it.

P2L26 '…scale (~ km2) and' Is this referring to 1 km2 to be considered large scale, or is a number missing?

It means "on a kilometer scale", and is considered to be large scale for PF, since spatial and temporal information on PF occurrence is usually only known on a plot scale (centimeters to meters).

P3L11 This section seems out of place, considering reorganizing with earlier paragraphs covering methods.

We will move this section to a different paragraph.

P6L1 Appendix A: consider presenting standard errors of the K measurements

We will add the standard error.

P8L15 How can observations at a single depth be considered sequential?

A sensor response in 10 cm only, is neither "no response" nor a "non-sequential response". Since we know that the upper sensor reacted first during a 10 cm only reaction (similar than during a sequential response), we included it to the category "sequential responses".

P11L5 I would start with the most interesting finding of this study, although it could be strictly seen as a result, I could see this information to be more suited in the methods section.

We will move the analysis of the rainfall event separation to the methods.

P24L10 The range of reported flow velocities both in this study and other reported studies generally seems extremely large. If the range is large to begin with, how is it remarkable that they fall in the same range? Perhaps I miss a part of the reasoning.

We will clarify the sentence. It is not remarkable that all these different soils fall in the same large range, but it is remarkable that they all show the same large variation. In other words there is no systematic difference of flow velocity with a certain type of soil and independent of the soil one can observe flow velocities from a few cm/day up to ~100000 cm/day.

P25L15 Awkward sentence structure.

We will rephrase the sentence.

P25L29 Although this seems like an insightful comment, are there any examples how this could be implemented, or is it readily tested on small scale? A reference would be useful here.

The hillslope parametrization of Loritz et al. 2017 (doi:10.5194/hess-21-1225-2017), which includes parametrization of fast flow path based on macropore field observations, could be seen as a recent example on a larger scale. The reference will be added.

---

## Author Response (AR1)

Response to comments of Referee #1 Heye Bogena

We thank Heye Bogena for reviewing our manuscript and for the helpful suggestions for improving the study. We answer below to each comment in a point-by-point reply. For clarity, the comments of the referee were copied in black and our response is in blue.

**General Comments**

Despite the novelty and interesting approach to studying preferential flow, I think this manuscript is not yet ready for publication and would benefit from a more clear description of methods (e.g. using flow charts) and from focusing its content to the most interesting parts.

The number of statistical analyses is rather excessive without providing much additional insights. For instance, the generalized linear regression model analysis is difficult to comprehend (especially for readers who are not familiar with this method) and does not provide clear results.

We have added a flow chart in the method section for a clear description of the analysis. Instead of analyzing small scale spatial patterns, we now focus on the temporal dynamics of soil moisture or rainfall characteristics of large scale spatial units (landscape units). We have removed information and statistical analysis in the text that is not supporting the main findings.

We have removed the generalized linear model (GLM) since it did not provide clear results and showed problems regarding the sample size and pseudo-replicates. As suggested by referee #2 and 3 we tested a generalized linear mixed effect model (GLMM) as an alternative (see response RC3). However, the results of the GLMM provide no new insights and since most variance is explained by the random spatial factors, the fitted GLMM results cannot be used for predictions in other areas. Hence, we did not include the GLMM into the revised manuscript.

Also the discussion section is too excessive should be focused on the most important results found in this study.

The result, discussion and conclusion section was reorganized and shortened to focus on the main findings.

A new aspect of this study is usage of calculated of water flow velocities on basis of 1D steady state flow assumptions to identify preferential flow events. However, although the velocity-based preferential flow assessment of events with a sequential order of sensor response times is appropriate, the derived matrix flow velocities may be prone to large errors. First, the steady state assumption is violated during infiltration events and second, the hydraulic parameters derived from field data which are influenced by preferential flow.

(1) Steady state assumption: During infiltration events a steady state is indeed not what we observe in nature. However, instead of a computationally intensive Richards based numerical 1D solution of all events, which also suffers from the uncertainty of parameters, the boundary conditions or discretization, we decided to use a steady state assumption of unsaturated flow. We tried to account for error that is based on the steady state assumption by using the maximum gradient during the event and are thus overestimating the driving forces. The hydraulic conductivity was previously calculated based on the median water content between the event begin and peak soil moisture. To be on the safe side, we have changed the calculation of the hydraulic conductivity to the maximum (peak) water content of the both sensors during the event. Hence, we overestimate matrix flow velocity rather than underestimating it, leading to a conservative estimate of preferential flow occurrence. Even though the absolute value is overestimated, it provides a maximum matrix flow velocity that can be used for the comparison to the magnitudes of measured flow velocities.

(2) Parameters: The parameters will not be completely unaffected by PF. However, as we wrote in the manuscript the parameters are derived on a daily timestep over many years (including dry phases) and thus do not include soil moisture dynamics on event base (e.g. hours). This prevents optimization of the parameters to be able to account for fast flow in the range of 1000 cm/hour. Therefore, the parameters were seen as a valuable alternative to pedotransfer functions for estimating matrix flow. Additionally, the potential influence of fast flow in the estimated retention parameters of Sprenger et al. (2016) will lead again rather to an overestimate of matrix flow velocities and thus underestimate the frequency of occurrence of preferential flow.

These circumstances may also explain why the measured NSR is lower for Marl grassland sites than predicted by the 1D steady state flow model.

For the hypothesis testing of NSR ($P_{int} > K_{mat}$) in the Marl grassland the hood infiltrometer $K_{mat}$ values were used (not the predicted values by the 1D steady state flow). We apologize for the unclear description and have clarified the methodology in the revised manuscript. Furthermore, we shortened the analysis and moved it to a different section of the results (section 3.1) to improve the structure of the revised manuscript.

Previous sensor based preferential studies (e.g. Wiekenkamp et al., 2016) used infiltrometer measurements to assign a meaningful threshold for the matrix flow velocity. Given that fact that hood infiltrometer measurements are available for the study area I suggest to use this data to define maximum matrix flow velocities for the different landscape units.

We changed the analysis and used both, modelled and measured matrix flow to differentiate between "slow" matrix and "fast" preferential flow.

**Specific comments:**

P3L5: Change to "...PF was more frequent during higher rainfall intensities.

We shortened the section and removed the sentence.

"P4L7: Change to "...and that therefore infiltration...

We rewrote the research questions and removed the sentence.

"P4L16: Add number of sites.

We added the number of sites to the section.

*"To test our research question we analyzed a dataset of 405 soil moisture sensors at 45 sites distributed across a complex landscape (varying geology and land cover) but under similar climatic conditions."*

P5L5: Change to "...mostly exhibit loamy texture."

We changed the sentence as suggested.

P5L5-6: Sentence reads awkward. Please reformulate and add landuse percentages.

We reformulated the sentence and land use percentage was added.

*"The land cover in the Luxembourgian part of the marl region is mainly characterized by agricultural sites (30 %) and grasslands (41 %, mainly pasture) with gentle slopes (~3°)."*

P5L6: How was the macroporosity defined/determined?

We specified this in the text.

*"The soils show high macroporosity documented by the excavation of horizontal soil profiles and counting of pores > 2 mm Ø."*

P5L15: Change to "METER Group Inc., USA"

We changed the sentence as suggested.

P6L8-9: You are actually calculating a "difference" and not a "macropore portion" of saturated hydraulic conductivity.

We agree that this term is misleading. Since this "macropore portion" of saturated hydraulic conductivity was only used for the GLM (which was removed) we removed the sentence.

P7L6: Why 6.7 mm?

The threshold was based on an intensity measurement of 80 mm/h ≈ 6.7 mm/5 min. Was changed to 80 mm h$^{-1}$.

P7L10-11: This should be reformulated in a less unconditional way, e.g. "The manufacturer gives an accuracy of..."

We changed the sentence as suggested.

P7L20: You should change "diel" into "diurnal" as this term is more common.

We changed the sentence as suggested.

P7L24-25: But later you decide to use 12 hour rainfall breaks, which is somewhat confusing. In order to make this more comprehensive I suggest to present the rainfall event delineation methodology completely in the method section (thus moving P11L5-15 to the method chapter).

Rainfall events were divided by 12-hour breaks, but soil moisture was tracked for additional 48 hours after the end of a rainfall event. However, we do agree that the different time steps for

event definitions are confusing and moved the mentioned section (P11 L5-15 and Table 2) to the methods and clarified some parts.

P8L6-8: This is not clear to me (e.g. what is the meaning of "0.4 %"?). Please be more specific.

We clarified the unit (0.4 Vol.%).

P9L08-15: There exists a vast literature on showing that preferential flow cannot be described with classical capillary theory. Why do you still pursue this analysis although this approach is obviously prone to fail?

Good question, however, models based on classical capillary theory are still very common. Anyhow, many people still have the misconception that preferential flow occurs only at saturation or at rainfall intensities exceeding infiltration capacity. We wanted to reiterate that this is not the case while at the same time the motivation was to see the how often classical theory fails, how strong the deviation is and if there are differences between different landscape units (probably some regions could be described by matrix flow).

However, to keep our main focus we have simplified and restructured the comparison of values predicted by capillary theory with measured values (see section 2.2.2 and 3.1).

 P9L11: The term "pore water pressure" only applies to subsurface water. Ponded water on the soil surface shows even positive pressures.

We agree that the sentence was unprecise. The section was shortened and the sentence was removed.

P9L13: better: "during a rainfall event infiltration capacity decreases as the soil approaches saturation"

We acknowledge the suggestion for an improved formulation. However, the analysis was simplified and the sentence was removed.

P9L27: Why should the aspect has an effect on the frequency of NSR occurrence?

Some studies found an effect of the slope aspect on the soil water properties (e.g Geroy et al. 2011, doi: 10.1002/hyp.8281). However, to focus on the main findings of our study this section was removed.

P10L4-5: better: ...as the velocity determined from the first responses of two sensors"

We changed the sentence as suggested.

P10L9-11: During infiltration events soil water flow should be governed by non-stationary conditions. Why do you believe that your stationarity assumption can be applied?

Please see our explanation under "General Comments".

P10L19-20: What does "in combination with parameter sets of Sprenger et al. (2016)" exactly means?

The formulation is confusing and was changed. van Genuchten equation was used and parametrized with the parameters from Sprenger et al. (2016).

*"For obtaining the matric potential the van Genuchten retention curves (van Genuchten, 1980) were parameterized using the parameter sets of Sprenger et al. (2016) (supplement Table S2)."*

P10L23-26: Nevertheless, as the inversely derived Ks-parameter of Sprenger et al.(2016) are derived from field data, they will still be affected by preferential flow and thus will be higher compared to a Ks derived from pure matrix flow.

Please see our explanation under "General Comments".

P11L22: Either use proportion or percentage

We acknowledge the suggestion for an improved formulation. However, we removed the sentence since it did not add important information.

P12L13: Add explanations of the abbreviations

We added an explanation of the abbreviations.

P13L18-26: This section comes somewhat out of the blue as it is not well related to the previous analysis. I suggest separating both sections and adding a short introduction to new one concerning soil water content changes.

We separated both sections and added a short introduction as suggested.

P13L20-21: This is difficult to understand. Please try to rephrase in a more comprehensive way.

We rephrased the sentence.

P15L2-3: This is an interesting finding. Does this correspond with seasonally varying precipitation properties?

The NSR pattern corresponds to the seasonal pattern in maximum precipitation intensity (highest from June – September) and soil moisture (lowest from July – October). We have added a graph to Figure 4 showing this.

P15L5-6: What are the possible reasons?

This result is discussed on P22L21-29. Higher macroporosity, stemflow or hydrophobicity are possible reasons. We restructured the discussion and pointed out possible mechanisms.

P16L6: The formulation "of up to~25 % of events" can be misinterpreted.

We changed the phrasing.

*"...(up to ~25 % of events)..."*

P16L10-11: Does this finding indicate that the inversely derived Ks-parameter of Sprenger et al. (2016) overestimate pure matrix flow?

Please see the explanation under "General Comments".

P17L1-17: I find this section not very meaningful as it cannot be well reproduced and the results are not very much enlightening. Therefore, for the sake of comprehensibility I suggest removing it.

Please see the explanation under "General Comments".

P18L11: It should "increasing" instead of "decreasing"

We apologized for this unclear phrasing. We meant "velocity is decreasing with decreasing water content." We shortened the section and changed the sentence.

*"There is no clear relationship of $v_{max}$ with $\theta_{ini}$ or $P_{max}$ and high maximum pore water velocities can be found over the full range of $\theta_{ini}$ and $P_{max}$."*

P18L15: I guess it should be again "increasing" instead of "decreasing"

Again we apologize for this phrasing and changed it accordingly.

P19L10: It would be interesting to see how choosing other rain gaps would influence the results (e.g. the proportion of NSR to SR events).

This was included in an earlier version of the manuscript but was removed since it made the results rather complicated and it did not add any additional information. The changes in the proportions (NSR/SR) of the reactions are relatively small (8% for Marl forest, ± 3-5% for all

other landscape units), increasing with longer rain gaps for most landscape units (comparing for example 6h, 12h, and 24h rain gaps). However, there is not a clear trend of increasing proportions with longer rain gaps for all landscape units. In general the number of rain events is decreasing with longer rain gaps and events last longer (see Table 2). This leads to a decrease of soil moisture events without a reaction (NR), while SR and NSR are increasing. However, the patterns between the landscape units stay similar.

P21L26: Did you compare pre-response analysis with entire event analysis? This would be interesting with respect to the comparison with the other studies.

Similar patterns are observed using total event rainfall amount or maximum rainfall intensity of the entire event. We added a sentence to give this information for comparison. However, since the response classification is not affected by the rainfall amount or intensity after the first soil moisture sensor response we keep the pre-response in our analysis and figures.

P21L28-29: I cannot follow this argument. Please explain in more detail.

In the sandstone grassland PF seems to be more often initialized at higher initial saturation, simply because infiltration capacity is lower and saturation is achieved faster compared to dry conditions. This is in contrast to the other landscape units with higher clay content, where more NSR is found under dry conditions with soil structure formation or hydrophobicity being the driving mechanism. The section in the discussion was restructured.

P23L7-11: In my view, the results of this study rather suggest that the occurrence of preferential flow is governed by unresolved small-scale structures and processes. The study of Wiekenkamp et al. (2016) used an even denser soil moisture sensor network and still could not find landscape properties to explain their results.

We totally agree. However, these small-scale structures and processes can probably be attributed to landscape properties. Different combinations of the landscape properties could lead to similar flow reactions making it hard to distinguish. We hypothesize that due to the high heterogeneity of soils it would need much more sensors (even more than in Wiekenkamp et al. (2016)) to identify them. Since we removed the analysis of landscape features, such as topography, the section was also removed.

P23L17-18: Why should the lower k_mat values of the Marl site lead to more NSR events and how do you know that the matrix infiltration capacity was underestimated?

See P9L13-15. The saturated matrix hydraulic conductivity ($K_{mat}$) was estimated using the hood infiltrometer that corresponds to the infiltration capacity at full saturation. We did not measure infiltration capacity at various moisture contents. Since the infiltration capacity is increasing with lower initial soil water content we rather underestimate the infiltration capacity under field conditions using $K_{mat}$ (because soils are rarely saturated in our catchment).

Our capillary-based estimation of NSR is again a conservative approach using this minimum infiltration capacity ($K_{mat}$). NSR is overestimated, because the infiltration capacity is underestimated (the threshold is more often exceeded then with a higher infiltration capacity).

However, to have a clearer structure of the study (especially results) and focus on the main analysis we will remove the comparison of the observed NSR responses with the estimated preferential flow reaction based on matrix hydraulic conductivity. Instead a short comparison of expected PF occurrence based on $P_{max}$ and $K_{mat}$ was added to section 3.1.

P23L18-19: Why should the overestimation of NSR by capillary theory in the Marl grassland be an indication of more vertical macropore flow?

We estimated more events with infiltration capacities (in our case $K_{mat}$) lower than maximum rainfall intensities in the Marl grassland and hence we should observe PF. Since we measured NSR less frequently than estimated by this approach, we hypothesized that these events have probably not resulted in PF with a NSR (non-homogenous flow), but rather in fast SR, which is supported by the high wetting front velocities. As correctly stated by the referee it has not to be vertical, but we do not observe a break in the sensor reaction sequence.

We clarified the possible mechanisms in the discussion (section 4.5) and supported the analysis with the seasonal $v_{max}$ patterns showing increased fast flow (high $v_{max}$) on grasslands during summer.

P24L23-28: These arguments are rather dubious.

We agree that these arguments are rather vague and speculative, hence the section was removed.

P24L23-P25L11: This section is a rather excessive discussion that does not provide much additional insights.

We partly removed this section and some sentences were moved to a different section. In general, the discussion will be restructured and will focus on the main findings.

P24L29: I guess it should be "increase" instead of "decrease".

We apologize again for causing confusion by our repeated erroneous phrasing.

P25L2-4: Remove repetition.

We removed the repetition.

P25L16: "...showed..."

The conclusion was partly rewritten.

P25L17: "...in deeper..."

The conclusion was partly rewritten.

P25L20: "showed" instead of "had"

The conclusion was partly rewritten.

P25L21: You did not prove the occurrence of "non-homogeneous wetter fronts". There are other mechanisms that can lead to preferential flow (e.g. by-pass flow).

We used NSR only as a proxy for preferential flow. However, we think that NSR in the first place only proved that there was a non-homogeneous wetting front that could be generated by preferential flow (see P8L22-23). This non-homogeneous wetting front can be generated by various preferential flow process (including by-pass flow). We have clarified that.

P25L23-P26L7: Please focus on presenting the main results in the conclusion section and avoid vague speculations.

We have rewritten the conclusion section and will focus on the main findings.

Figures

Figure 9: Dots are difficult to discern.

The figure was removed.

**Response to comments of anonymous Referee #2**

We thank the anonymous referee #2 for reviewing our manuscript and the comments concerning mainly statistical issues. We answer below to each comment in a point-by-point reply. For clarity, the comments of the referee were copied in black and our comments are in blue.

**General Comments**

The results are very difficult to follow and it is not always clear why certain analyses were done and how they link back to the main objectives of the study. It is often not clear how and why soil profiles were grouped for certain analyses. I encourage the authors to provide more clarity on the analyses and highlight how the analyses address the research objectives.

We added a flow chart in the method section for a clear description of the analysis and clarified the grouping of the soil moisture profiles for each analysis step. We have also better connected the different sections to the objectives and therefore improved structure and readability.

a) Statistical issues

Some of the key conclusions made in this study rely on frequentist statistical testing (e.g. p-values), which, as the authors acknowledge (p23,l5-11), can be highly sensitive to sample size issues. There has also been considerable discussion recently about the major limitations to this approach (see Amrhein et al. 2019. Nature 567:305-307 for a very recent example, also Wasserstein and Lazar 2016. The American Statistician 70:129-133). It might be valuable if the authors discuss some of the inferential uncertainties and limitations of their approach.

It is true that different sample sizes can lead to problems in the interpretation of test statistics (as shown in e.g. Amrhein et al. 2019. Nature 567:305-307) and that the p-value should not be used as a rigorous yes/no criterion. However, test statistics can give additional insights, especially while comparing large samples. None of our analysis and interpretations is purely based on p-value statistics and they were just added to give additional information (such as error bars, distributions etc.). Since sample size is a critical issue we added the sample size where it was missing in the manuscript (or supplement) to support interpretation.

1) pseudo-replication: It seems to me that the statistical models should be fit to the 45 sites, not the 135 soil profiles, since the grouping of three profiles within each site cannot be treated as independent. Focusing on the 135 soil profiles could be done within the GLMs if within-site variability were accounted for, but it's not clear to me that this was done.

The linear model (LM) was fitted to the mean NSR percentage of the 45 sites. The generalized linear model (GLM) was fitted to the infiltration events of the 135 profiles since it covers the temporal domain. Indeed pseudo-replications are an issue for the GLM. We tested a generalized linear mixed effect model (GLMM) as an alternative (see response RC3). By using a GLMM for the 45 sites with the individual sites as the random effect of the model we avoided pseudo-replications and treated the individual spatial landscape effects (geology, land-cover, slope, aspect etc.) together as one random landscape effect.

However, the results of the GLMM provide no new insights and since most variance is explained by the random spatial factors, the fitted GLMM results cannot be used for predictions in other areas. Hence, we did not include the GLMM into the manuscript.

2) sample size vs number of predictor variables used in the models: Although there is an impressive amount of data collected for this study, I'm concerned that some of the results (e.g., identification of statistically significant predictors) are simply the product of small sample sizes and noise in the model fits. For example, the GLM for Grassland- Sandstone was fit to 9 soil moisture profiles (so really, just three sites), but 13 predictor variables were used in the model fitting, which will result in an underdetermined solution. If the authors decide to keep the statistical analyses, I would suggest some sort of cross-validation exercise be done to assess the rigor of the models.

The GLM was fitted to all events of the 9 profiles of the Sandstone grassland sites, which are 698 data points. Furthermore, some predictors were removed by stepwise AIC, so that only four predictors were used (not 13). Hence, the model does not result in an underdetermined solution. However, as already mentioned above, the GLM was removed from the analysis.

3) data exclusion. It was suggested in the methods that there is some incompleteness to the time series for each soil profile (due to logger failures and criteria for including data in the analysis). How many sites and profiles were excluded and for what time periods? This is important to know as it relates to the sample size issue outlined above.

We added a diagram to the supplement that shows how many profiles in which landscape units were "active" (met the quality criteria) over the entire time period (~2012-2017).

b) Within-site and temporal variability

Instead of focusing on statistical significance, I think the authors could make an excellent contribution by focusing more on the within-site and temporal variability of their field measurements. My understanding is that the grouping of the sampling approach can be organized as: geology - land cover - site - profile. Most of the analysis focuses at the geological and land cover levels; however, throughout the manuscript I found myself constantly wanting to know more about the within-site variability in terms of both infiltration event characteristics and soil properties. Also, at the profile level, I wanted to know more about the temporal variability. Did profiles that exhibit NSR only exhibit NSR or did they shift between NR, SR and NSR? If so, why? Instead of generalizing the results using p-values, I suggest focusing on graphical approaches to show evidence to support the research objectives.

The within-site/profile variability is indeed an interesting topic. We will add a sentence about the within-site variability to the results. Table 3 already gives some information about the within-profile variability.

*"The NSR variability between the single profiles within a landscape unit was found to be high (Table 3). The site-intern variability of NSR (profiles within the same sites) measured as the median standard deviation was highest in marl (forest: 7.5 %, grassland 6.4 %) followed by slate (forest: 4.2 %, grassland 6.1 %) and sandstone (forest: 1.9 %, grassland 3.0 %)."*

However, the site or profile-level variability was not the main aim of this study. Other studies have already focused on that topic (see e.g. Wiekenkamp et al. (2016) or Liu and Lin (2015) in the reference list of the manuscript). The aim was to show the effect and variation of larger-scale landscape units with different properties. Furthermore, we wanted to identify potential temporal differences and similarities among their reactions. We clarified the aim of the study in the revised manuscript.

Many results in the manuscript include graphical approaches (see Fig. 3, 4, 6, 7, 8 and new diagrams were added to the revised manuscript). We do not think that adding the test statistics weakens our findings (see response to General Comments).

**Specific Comments**

p1,l27-28: Consider incorporating the parenthetical into the sentence - as is, this makes for a weak opening.

The parenthetical was removed.

p2,l17: Consider removing this last sentence or expand on it to clarify to the reader what is meant by hotspots and hot moments of PF.

We removed the sentence.

p2,l29-p3,l18: Consider revising these paragraphs. Right now these feel like simply a list of results from other studies. I suggest trying to better synthesize these results and identify key findings and knowledge gaps.

We have revise the paragraph and summarized the studies.

p4,l9-11: I think the research questions could be improved. What is meant by 'underlying controls'? Has this actually been done in this study? It seems like the PF proxies are linked to precipitation, landscape, and soil characteristics through statistical modeling. 'Underlying controls' suggests to me a more process-based approach (e.g., soil physics modelling), which isn't done in this study - outside of the predicted matrix flow velocities. What is meant by temporally stable?

Indeed we did not clearly identify processes. The research question will be rephrased. By "underlying controls" we meant spatial and temporal influences or drivers of preferential flow occurrence on a larger scale (e.g. landscape units).

"Temporally stable" refers to the preferential flow occurrence (if it is changing over time or not).

*"Therefore, the main aim of this study is to identify and compare the temporal dynamic of PF occurrence by using profiles of soil moisture sensors in different large-scale spatial units that could potentially be used as representative units for catchment modelling. Since it can be expected that rainfall intensity and soil moisture have a strong influence on the initialization of PF (Beven and Germann, 1982) we will mainly focus on the temporal controls of initial soil moisture and rainfall. More specifically, we attempt to answer the following question: Does PF occurrence increase with rainfall intensity since higher intensity leads more frequently to an exceedance of matrix infiltration capacity? Does PF occur more often under wet conditions*

*since the infiltration capacity is lower? How is the temporal PF dynamic influenced by spatial factors like geology/soil type and land cover?"*

p5,l25-26: What is the orifice diameter of the rainfall gauges? How was the placement of the forest gauges determined? Was variability in canopy cover and throughfall a concern?

The orifice diameter of the rain gauges is 16.5 cm (collection area 214 cm²). The rain gauges were randomly placed on the 29 forest sites. The information was added to the sentence. The experimental design of placing the five throughfall gauges aimed at covering the variability in canopy cover at each site, and variability in measured throughfall between the gauges was expected.

p6,l3: Why weren't infiltrometer measurements available for the grassland/Sandstone sites?

Hood infiltrometer measurements are often time consuming and we were not able to measure all sites during the same field campaigns.

p7,l5: Why would the sensors log these kinds of 'implausible' events? How many events were rejected because of these criteria?

During the reconnecting of the loggers following a logger error (no power etc.), the rain gauges sometimes produced this kind of implausible events. Furthermore, clogging and release of the clogged water could be a reason of the unrealistic rainfall events. The number of rejected events was added (text and flow chart).

*"These implausible events were observed to happen during the reconnecting of the loggers following a logger error (no power etc.) or clogging and release of the clogged water."*

*"By applying the quality criteria for rainfall events using $t_e$ = 12 h, 1392 of 32025 rain events (sum of profile rainfall events) were excluded because of the threshold criteria and 426 because the mean temperature was below 0°C during the event."*

p7,l29-31: How many times were data from a profile rejected because of these criteria?

We included the number of rejected soil moisture event and the explanation into the section and the flow chart.

*"From the total of 30207 rainfall events, 15645 could be used for the analysis of the soil moisture, since they allowed for a clear separation of soil water flow by more than 24h without a new rainfall input. 7395 of these events did not meet the quality criteria of completeness and consistency of the soil moisture time series, hence 8250 infiltration events (sum of soil moisture*

*event observations at all 135 profiles) could be used for the analysis. Changing the completeness criterion from 99% usable soil moisture data points during an event to e.g. 95% is only slightly affecting the number of infiltration events (e.g. 8353 events usable in the analysis). This is due to the fact that most exclusions result from long term failure of one sensor of a profile that leads to a complete exclusion of the entire profile."*

Table 1: The first row highlights to me the potential issue of pseudo-replication in this study. It seems more appropriate to report the number of sites, not profiles. Also, for the soil texture and mean clay content, how variable were these values within geological and cover class combinations?

Table 1 is just an overview of the study sites. The different sensor responses (SR, NSR) were calculated for every single soil moisture profile. We think it is appropriate to give the number of profiles, since they determine the number of observations. We added the full textural information and the standard deviation to the supplement materials.

p9,l18-26: This paragraph is unclear. Why were some GLMs fit for individual landscape units and one GLM fit to all profiles? What was the sample size used in each of these models? If I understand this correctly, the GLM model for Sandstone-Grassland was fit to just 3 sites (9 profiles)? This seems like much too small sample size to fit models with up to 13 predictor variables.

Please see our response under General Comments. The GLM was fitted to the individual landscape units to test for differences in the predictors on this scale. Furthermore, we compared it with one GLM for the whole catchment to see the potential for such an approach.

However, as already mentioned above, the GLM was removed from the analysis.

p9,l27: Why restrict to slopes > 10%?

A slope < 10% is relatively flat and the orientation is not strongly pronounced. To focus on the main findings of this study (temporal variability of PF occurrence between landscape units), the analysis of the aspect was removed.

Fig 2a: I'm not sure what conclusions to draw from this graph? The text suggests that it shows no clear difference between landscape units; however, there appear to be considerable differences in cumulative precipitation (e.g., for event numbers ~ 150, we see a range of P[sum] of almost 750 mm). Also, I don't think the various landscape units share the same event records (i.e., the x-axis is not sequential for each landscape unit), so are they comparable?

It is correct that the events are not identical and necessarily sequential due to the rainfall heterogeneity, quality criteria, and the length of the time series. Therefore, single sites can show high deviation even within the same landscape unit. The motivation was to show that there is no systematic difference between the landscape units. We think that Fig. 2b) provides enough information and we will remove Fig. 2a).

p11,l24: Why only 2014 and 2015?

These were the first two years with all sites installed. Since the calculated proportions of these two years (and also the other years) do not add additional relevant information for interpretation, we removed the section to focus on the main findings.

p15,l1: Would a relationship between NSR and distance to stream be expected? Perhaps a provide a rationale.

Some authors found a relationship on hillslope position (see references P22L32-P23L4). We agree that an explanation should be mentioned earlier in the manuscript and is currently missing. However, the analysis of the small scale spatial patterns will be removed to focus on the main findings (temporal patterns of soil moisture and rainfall).

Fig 4: Where do these data come from? Are these averages across all years in the study? How much did this vary between years? Does this figure account for differing number of events each year or exclusion of profiles due to logger failure or selection criteria?

Fig. 4 is based on the same data as all other diagrams. It shows the mean NSR of all events that were measured in the twelve individual months independent of the landscape unit. Hence, the diagram averages across different years. We have modified this diagram and separate between forest and grassland sites and added additional shaded areas to show the variability among years. The number of events used for this analysis (number of events per month for all years and the min. and max. of individual years) was added to a table in the supplement.

Fig 8: The fit is statistically significant (but see general comment (a) above), but is this relationship practically meaningful? If you were to remove the line of best fit, I'm not sure someone would identify a relationship in these data.

We totally agree, the fit is statistically significant, but explains only little variation (we wrote this on P18L11-12). Furthermore, on P24L28-30 we note: "The $\theta$-$v_{max}$ relationship shows that even though the decrease of $v_{max}$ with [decreasing] $\theta$ is significant, it has little explanatory power and fast flow ($>1000$ cm day$^{-1}$) can occur at any $\theta$." The fit was included to highlight the

strong variation that not simply follows the trend. However, we removed the fit in the new version of the manuscript.

p22,l32: How much range in hillslope position was sampled? Was this the distance to stream metric?

The range of hillslope position was determined by the distance to stream with a range between 4 and 251 m from the different sites to the stream. Please see the table in Appendix A. However, this analysis was removed to focus on the main findings.

p24,l10: Speculations on why this is?

Texture seems not to be the main driver of water flow velocity during infiltration in the classical manner that fine grained texture corresponds to slow flow. Infiltration seems to be strongly controlled by PF phenomena, which are dependent on soil structure (influenced by a high clay content), biotic macropores (roots channels, earthworm borrows) and initiation processes (hydrophobicity, rain intensity). The high heterogeneity of the landscape and its temporal variation leads to PF that is caused by different drivers that are partly independent of texture (e.g. organic carbon content, number and species of soil organisms, vegetation type, rainfall characteristics). We clarified this in the conclusion of the revised manuscript.

**Response to comments of anonymous Referee #3**

We thank the anonymous referee #3 for reviewing our manuscript and his suggestion for improving the temporal occurrence model of preferential flow (PF). We answer below to each comment in a point-by-point reply. For clarity, the comments of the referee were copied in black and our comments are in blue.

**General Comments**

Primarily the structure and selection of results should be reconsidered, but also a more defined storyline could assist the reader to extract the main novelties of this study. In general, the manuscript could benefit from reconsidering what information is necessary to broadcast the main message. I recommend to consider a few key figures that conveniently show the reader the approach and main interesting findings, instead of a long sequence of tables and graphs. Lastly, the readability would greatly increase if the authors consider a key phrase in each paragraph that, perhaps almost trivially, highlights what should be learned from the given information.

For the structure of the paper, I would recommend to consider separation of the hypothesis and throughout the paper clearly indicate which sections address information for which hypothesis. I miss this in the paper. The hypothesis could possibly be broken up in two sections. For example: 1) PF is the dominant process during infiltration, and 2) capillary theory does not suffice to explain infiltration. These can be tested for the given explanatory factors, such as land cover, geology etc. Which also gives more structure in the result and discussion section.

In the revised manuscript we have restructured the methods and results and focused on the main findings, the temporal dynamics of PF in different large scale spatial units (landscape units). By removing most of the small-scale spatial analysis we highlighted the main storyline. Additionally, we added introduction phrases for the single sections. A flow chart further helps to follow the analysis that were performed. The comparison with the capillary theory, as one of the aims of the study, was removed.

We think that these changes help to improve the readability and to follow the storyline of the manuscript.

The generalized linear model (GLM) provided insight in the explanatory power of a large set of variables. However, as anonymous referee #2 addressed, there are some limitations to this approach. I will not re-evaluate these points, but I instead would recommend the authors to consider the use of mixed effect models. This approach allows to include random factors that potentially explain variability but are not directly incorporated in the study design. Seen the authors use R, the packages 'lme4', 'lmerTest', and 'nlme' could relatively easily allow to explore the use of mixed effect models.

We thank you for the suggestion and removed the GLM and tested a generalized linear mixed effect model (GLMM) that incorporates the spatial site information as a random effect.

We fitted a binomial GLMM (using the R package lme4) to the response classification of all our 8250 infiltration events with a logit link function due to the binary nature of our data (*NSR* yes/no). The spatial domain was taken as the random effect (random intercept and slope) on the scale of the 45 sites. For evaluating the model, the $R^2$ (delta method) for GLMMs introduced by of Nakagawa et al. 2017 (http://dx.doi.org/10.1098/rsif.2017.0213) was calculated using its implementation into the "R" package "MuMIn" (https://cran.r-project.org/web/packages/MuMIn/index.html). The $R^2$ of a GLMM can be divided into a marginal $R^2$ ($R^2_m$), which gives the proportion of explained total variance by the fixed effects and a conditional $R^2$ ($R^2_c$) that gives the explained variance of both fixed and random effects. We found a $R^2_c$ of 0.17 and a $R^2_m$ of 0.03 showing that most variance is explained by the random effects.

The results of the GLMM provide no new insights and since most variance is explained by the random spatial factors, the fitted GLMM results cannot be used for predictions in other areas. Hence, we did not include the GLMM into the manuscript.

On a final note, I wonder if there is any indication that the contributing area of each site is independent of the occurrence of NSR? A correlation could guide towards rising groundwater tables and associated capillary rise, or horizontal flow. Especially with high antecedent soil moisture groundwater response could be relatively fast when contributing area is large.

We have calculated the upslope contributing area for each site and compared it against NSR occurrence. The Spearman R is 0.1 and hence, influence of groundwater in the upper 0.5 m of soil seems to be small for our sites. This is supported by the fact that % NSR and distance to stream does not show a correlation (P15L1).

**Specific Comments**

P2L17 Seems out of context to mention hotspots or hot moments, especially as a final statement of the section. The statement needs further elaboration and references.

We agree that the sentence is out of context at the end of this section and removed it.

P2L26 '…scale (~ km2) and' Is this referring to 1 km2 to be considered large scale, or is a number missing?

It means "on a kilometer scale", and is considered to be large scale for PF, since spatial and temporal information on PF occurrence is usually only known on a plot scale (centimeters to meters).

P3L11 This section seems out of place, considering reorganizing with earlier paragraphs covering methods.

The section was reorganized and shortened.

P6L1 Appendix A: consider presenting standard errors of the K measurements

We have added the standard error.

P8L15 How can observations at a single depth be considered sequential?

We changed the classification and combined the 10 cm only reaction together with the NR events (no response) to the new class of "not classifiable" infiltration events (NC).

P11L5 I would start with the most interesting finding of this study, although it could be strictly seen as a result, I could see this information to be more suited in the methods section.

We moved the analysis of the rainfall event separation to the methods.

P24L10 The range of reported flow velocities both in this study and other reported studies generally seems extremely large. If the range is large to begin with, how is it remarkable that they fall in the same range? Perhaps I miss a part of the reasoning.

We have clarified the section.

*"In summary, it is remarkable that no clear differences in flow velocities between different soil types could be identified (neither in our study nor across all previous studies). Instead, all soil types showed a similarly large range of velocities ($10^0 – 10^5$ cm day$^{-1}$). Furthermore, one can*

*see orders of magnitude difference in $v_{max}$ between different events but not among the landscape units.''*

P25L15 Awkward sentence structure.

The conclusion was partly rewritten.

P25L29 Although this seems like an insightful comment, are there any examples how this could be implemented, or is it readily tested on small scale? A reference would be useful here.

The conclusion was partly rewritten and the sentence was changed.

Response to comments of Referee #4 Nicholas Jarvis
* * *
We thank Nicholas Jarvis for reviewing our manuscript and his comments on the preferential flow phenomena we have observed. We answer below to each comment in a point-by-point reply. For clarity, the comments of the referee were copied in black and our comments are in blue.

**General Comments**

As the fourth person to comment, I hesitate to add too much to what the others have already written. However, one thing that surprised me was the data on hydraulic conductivity measured by Hood infiltrometer. Some of the values at matrix saturation are as large as 500 cm/day, which seems excessively large for infiltration rates measured at a tension of 6 cm, ostensibly unaffected by soil macropores. Even some of the total saturated hydraulic conductivities seem extraordinarily large to me, varying up to 1500 to 2000 cm/day. Maybe my surprise is just a consequence of the fact that I am more familiar with arable soils, not forest soils. No details are given of the method. I wonder, for example, how the 3D nature of the flow under the infiltrometer is accounted for? If it isn't accounted for, you could seriously overestimate K, especially in strongly layered soils. Could the authors give more details on the method?

Forest topsoils can have extremely high saturated hydraulic conductivities (see e.g. Greenwood & Buttle 2014, doi: 10.1002/eco.1320; Gonzalez-Sosa et al. 2010, doi: 10.1002/hyp.7640). Our soils were very structured and permeable (sometimes infiltration was too high to fill the hood of the infiltrometer). At some points the values were verified with double ring infiltrometer measurements being in the same range of conductivities. All measurements include tensions close to saturation and hence, the saturated hydraulic conductivity values (tension 0 cm) are more reliable. Matrix saturated hydraulic conductivity (tension 6 cm) was calculated from a Gardner function that was fitted to the measured tensions. Due to the high macroporosity at many forest locations pressure in the hood was difficult to adjust and measurements could only be conducted for maximum tensions of 1-3 cm. Hence, fore some sites matrix saturated hydraulic conductivity is just an extrapolation of the Gardner fit. However, high matrix saturated hydraulic conductivities were mainly measured in the sandy topsoils of the Marl and Luxemburg Sandstone and therefore the values seem to be plausible to us.

The hood infiltrometer is described in greater detail in Schwärzel & Punzel 2007 (doi: 10.2136/sssaj2006.0104). The derivation of matrix saturated hydraulic conductivity from

measured infiltration rates (hood infiltrometer) accounts for the 3D nature of flow using the solution of Woodings 1968 (steady state infiltration from a circular source).

We have clarified the hood infiltrometer method in the revised manuscript.

The authors attempt to test the hypothesis that preferential flow in macropores in only generated if the rainfall rate exceeds the matrix infiltration capacity, such that the pore water pressure is close to atmospheric pressure at the soil surface. But their approach is rather indirect and therefore prone to errors and uncertainties. The best (only proper?) way to test this hypothesis would be to install tensiometers to measure soil water pressure potential, as well as the probes for soil moisture content. I think their conclusions on this point may be a little suspect, especially considering the unusually large matrix infiltration rates they measured (see above).

We agree that tensiometers would help to validate our preferential flow observation. At each site we had one profile of the Decagon MPS-2 sensors which measure only water potentials < -90 hPa. Therefore the sensors were not suited to detect preferential flow.

Our method (max. rainfall rate exceeds the matrix infiltration capacity) is indeed an indirect estimation of preferential flow occurrence, as mentioned by the referee. In contrast to a direct method (like tensiometers) the aim of the analysis was not to validate our preferential flow measurements, but rather to compare the observations with an estimation by a capillary approach (based on matrix hydraulic conductivities).

To have a clearer structure of the study (especially results) and focus on the main analysis we have removed this comparison of measured data with a capillary approach ($P_{max} > K_{mat}$) to a large extend.

Connected to this, I think the authors could consider re-phrasing the text at lines 739-743: these non-capillary flow mechanisms certainly contribute to flow close to saturation. However, studies of the physics of these flow processes suggests that they also require pressures quite close to atmospheric pressure for them to generate faster flow velocities than those in the matrix (see discussion and cited papers in Jarvis, 2007, p.528-529). I haven't seen any later studies that clearly contradict those findings.

It is correct that also alternative flow processes (e.g. film flow) require a relatively low soil water potential (high saturation). On P23L13-15 we wrote: "Higher occurrence of measured

*NSR* compared to capillary theory prediction could indicate other initiation and flow mechanisms [...]". The sentence is vaguely phrased and we apologize for that. The meaning was, that unknown initiation processes (local depressions, channeling of water by vegetation, hydrophobicity, etc.) can locally lead to higher water contents and alternative flow processes. We have clarified this sentence.

*"[...] Furthermore, the mismatch of measured PF occurrence (NSR, fast $v_{max}$) compared to the prediction based on $P_{max}$ exceeding $K_{mat}$ indicates that initiation processes such as hydrophobicity/water repellency, local microtopographic depressions or channeling of water by vegetation could be the reason of the frequent occurrence of PF (Blume et al., 2008; Doerr et al., 2000; Schwärzel et al., 2012; Weiler and Naef, 2003). Locally, these processes can lead to higher water contents and thereby pressures at the soil surface close to atmospheric pressure which in turn trigger PF."*

The fact that preferential flow is strongest when the soil is dry suggests that the likeliest explanation of your results is the occurrence of water repellency, which is known to be a common feature of forest soils. Water repellency causes water potentials to quickly reach very close to zero, even during quite light rainfall, so that water can flow into surface-vented macropores even when the soil is dry. The authors do briefly mention hydrophobicity as a possible reason for their results (lines 694-697), but then seem to dismiss it, which I think is a pity. Preferential flow through macropores generated by the occurrence of (sub-critical) water repellency has been reported in several studies in recent years (see those cited in the review by Jarvis et al., 2016. Vadose Zone Journal, doi:10.2136/vzj2016). I think this topic should be discussed more fully in the paper and some of these recent studies cited.

We agree. We have restructured and partly rewritten the discussion and have stronger considered hydrophobicity as an initiation mechanism of PF.

I didn't get a clear idea of whether the hypothesis on Line 145 was accepted or rejected? The first question is what is meant by "dominate"? Is it the frequency of rain events that generate preferential flow or the amount of water recharging through the unsaturated zone (or something else)? Looking at the text on lines 426 and 432-433, it would seem that preferential flow was not a dominant process (which would also tally with the very high matrix saturated hydraulic

conductivities). But I got a different impression from the conclusions, at lines 828-832. Could this be clarified?

The word "dominates" might be too strong and we have rephrased the research question and the conclusion. We mainly focus on the frequency of preferential flow occurrence. To draw a conclusion on the amount of water that is transported or that contributes to groundwater recharge it would require a physically based model (out of the scope of this study). Therefore, we only used the observed water content change as an estimate.

To answer the question if preferential flow is "dominant": We found preferential flow in all our landscape units, but being temporally highly variable. We were able to find hotspot landscapes (clayey soils, forests) and hot moments (dry, high rainfall intensity) of preferential flow occurrence. This verifies that preferential flow is a common and important, spatially and temporally variable process, but maybe not a dominating process.

New research question and aim:

*"Therefore, the main aim of this study is to identify and compare the temporal dynamic of PF occurrence by using profiles of soil moisture sensors in different large-scale spatial units that could potentially be used as representative units for catchment modelling. Since it can be expected that rainfall intensity and soil moisture have a strong influence on the initialization of PF (Beven and Germann, 1982) we will mainly focus on the temporal controls of initial soil moisture and rainfall. More specifically, we attempt to answer the following question: Does PF occurrence increase with rainfall intensity since higher intensity leads more frequently to an exceedance of matrix infiltration capacity? Does PF occur more often under wet conditions since the infiltration capacity is lower? How is the temporal PF dynamic influenced by spatial factors like geology/soil type and land cover? "*

Finally, one general comment on terminology: I think it would good if the authors avoided the use of the term "wetting front" and "wetting front velocity". If you have strong preferential flow, there should not be a well-defined wetting front. Maybe you can write "maximum pore water velocity" instead of "wetting front velocity"?

We agree that the term is not precise. We used this term since it is relatively often used in the literature (see e.g. Hardie et al. 2013, doi: 10.1016/j.jconhyd.2012.10.008; Germann & Hensel 2006, doi: 10.2136/vzj2005.0080). However, the term "maximum pore water velocity" is more appropriate and we have changed it according to your suggestion.

**Specific Comments**

1.) Line 41: Jarvis (2016) is not in the reference list. I think you mean Jarvis et al. 2016?

We apologize for giving a citation that is not in the reference list. The citation was changed to the intended reference: Larsbo et al. 2014 (doi:10.5194/hess-18-5255-2014).

2.) Lines 64-68: you neglected one very important method and that is the analysis of breakthrough curves for non-reactive solutes (tracers). Perhaps this could be added here with one or two appropriate references?

We have added the analysis of breakthrough curves as a potential method with Koestel et al. 2013 (doi:10.1002/wrcr.20079) as a reference.

3.) Lines 70-72: These are not really direct measurements (see line 69). In this respect, X-ray tomography of flow/transport is the only method that gives direct measurements (see Sammartino, S., et al. 2015. Identifying the functional macropore network related to preferential flow in structured soils. Vadose Zone J., doi:10.2136/vzj2015.05.0070; Koestel, J., Larsbo, M. 2014. Imaging and quantification of preferential solute transport in soil macropores. Water Resources Research, 50, 4357–4378).

We agree that the mentioned methods are no direct measurements. We will change the sentence to: *"Another way to identify the potential for PF are measurements that can be related to the number and volume of macropores or cracks."*

4.) Line 202: robur

We corrected the latin name.

5.) Line 311: "non-uniform flow" is simpler and better than "non-homogeneous wetting front"

We changed it as suggested to the term "non-uniform flow".

6.) Lines 324-328: This is confusing. I think it could be written more clearly and much simpler: "In addition, the hypothesis is tested that preferential flow in macropores is only generated if the rainfall rate exceeds the matrix infiltration capacity, such that the pore water pressure reaches values close to atmospheric pressure at the soil surface"

We thank you for this suggestion of an alternative and simpler phrasing. However, the sentence was removed due to the restructuring of the methods section.

7.) Line 368: "matric" not "matrix"

We corrected the word as suggested.

8.) Line 377: Delete "Mualem"

We removed the word "Mualem".

[revised manuscript text omitted]

S̶a̶n̶d̶e̶r̶,̶ ̶T̶Rye, C. F̶.̶ and G̶e̶r̶k̶e̶,̶ ̶H̶.̶ ̶H̶.̶:̶ ̶P̶r̶e̶f̶e̶r̶e̶n̶t̶i̶a̶l̶ ̶f̶l̶o̶w̶ ̶p̶a̶t̶t̶e̶r̶n̶s̶Smettem, K. R. J.: Seasonal and Interannual Variability of the Effective Flow Cross-Sectional Area in p̶a̶d̶d̶y̶ ̶f̶i̶e̶l̶d̶s̶ ̶u̶s̶i̶n̶g̶ a d̶y̶e̶ ̶t̶r̶a̶c̶e̶r̶Water-Repellent Soil, Vadose Zo. J., 6̶(̶1̶)̶,̶ ̶1̶0̶5̶–̶1̶1̶5̶14(3), 0, doi:10.2136/v̶z̶j̶2006.0035, 2̶0̶0̶7̶.̶

v̶a̶n̶ ̶S̶c̶h̶a̶i̶k̶,̶ ̶N̶.̶ ̶L̶.̶ ̶M̶.̶ ̶B̶.̶:̶ ̶S̶p̶a̶t̶i̶a̶l̶ ̶v̶a̶r̶i̶a̶b̶i̶l̶i̶t̶y̶ ̶o̶f̶ ̶i̶n̶f̶i̶l̶t̶r̶a̶t̶i̶o̶n̶ ̶p̶a̶t̶t̶e̶r̶n̶s̶ ̶r̶e̶l̶a̶t̶e̶d̶ ̶t̶o̶ ̶s̶i̶t̶e̶ ̶c̶h̶a̶r̶a̶c̶t̶e̶r̶i̶s̶t̶i̶c̶s̶ ̶i̶n̶ ̶a̶ ̶s̶e̶m̶i̶ ̶a̶r̶i̶d̶ ̶w̶a̶t̶e̶r̶s̶h̶e̶d̶,̶ C̶a̶t̶e̶n̶a̶,̶ ̶7̶8̶(̶1̶)̶,̶ ̶3̶6̶–̶4̶7̶,̶ ̶d̶o̶i̶:̶vzj2014.10.1̶0̶1̶6̶/̶j̶.̶c̶a̶t̶e̶n̶a̶.̶2̶0̶0̶9̶.̶0̶2̶.̶0̶1̶7̶,̶ ̶2̶0̶0̶9̶0141, 2015.

[revised manuscript text omitted]


Appendix B: Generalized linear regression models and linear model

Full equations for the generalized linear model (*GLM*) for estimating temporal occurrence for a *NSR* and the linear model to predict the spatial occurrence of *NSR*.

| GLM - Probability of event NSR | | Intercept | $P_{sum}$ [mm] | $P_{max}$ [mm h$^{-1}$] | $P_{int}$ [mm h$^{-1}$] | $\theta_{ini}$ [-] | elevation [m a.s.l.] | aspect [°] | slope [°] | distance to stream [m] | distance to tree [m] | rooting depth [cm] | Avg. Monthly air temprature [°C] | $K_{mat}$ [cm day$^{-1}$] | $K_{MP}$ [cm day$^{-1}$] | McFadden Pseudo-R² |
|---|---|---|---|---|---|---|---|---|---|---|---|---|---|---|---|---|
| 5 groups | | -2.452* | 0.023* | 0.025* | - | -3.851* | 0.003* | -0.001 | - | -0.003* | X | -0.007* | - | - | 0.001* | 0.08 |
| Forest | Slate | -2.784 | 0.02* | 0.036* | - | -10.97* | 0.008 | - | -0.027 | - | - | -0.005 | - | - | - | 0.09 |
| | Marl | -0.538 | 0.059* | 0.042* | - | -7.172* | - | - | - | -0.062* | 0.253 | - | - | - | - | 0.18 |
| | Sandstone | 4.386 | 0.035* | - | - | -14.983* | - | 0.015* | - | -0.016* | - | -0.11* | - | -0.102* | 0.062* | 0.21 |
| Grassland | Slate | -47.859* | 0.034* | 0.03* | - | 3.079 | 0.118* | - | -0.211* | -0.056* | X | -0.099 | - | - | - | 0.16 |
| | Marl | -3.351* | - | 0.051* | -0.253 | 9.078* | - | -0.012* | -0.538 | - | X | - | - | - | - | 0.10 |
| | Sandstone | 25.056 | 0.042* | - | - | - | -0.092 | - | - | - | X | - | -0.136* | X | X | 0.09 |

| Linear model - % NSR on profile scale | Intercept | Median profile $P_{sum}$ [mm] | Median profile $P_{max}$ [mm h$^{-1}$] | Median profile $P_{int}$ [mm h$^{-1}$] | Median $\theta_{ini}$ [-] | elevation [m a.s.l.] | aspect [°] | slope [°] | distance to stream [m] | X | rooting depth [cm] | Avg. Monthly air temprature [°C] | $K_{mat}$ [cm day$^{-1}$] | $K_{MP}$ [cm day$^{-1}$] | R² |
|---|---|---|---|---|---|---|---|---|---|---|---|---|---|---|---|
| 5 groups | -0.2996 | 0.0737* | - | -0.5063* | 0.5081* | 0.0004 | - | - | -0.0001 | X | -0.0009 | - | -0.0002 | 0.0001* | 0.40 |

- : Not required predictor (stepwise AIC)          *  p < 0.01